# QUANTSPARSE: COMPREHENSIVELY COMPRESSING VIDEO DIFFUSION TRANSFORMER WITH MODEL QUANTIZATION AND ATTENTION SPARSIFICATION

**Weilun Feng**[1,2], **Chuanguang Yang**[1,]*  **Haotong Qin**[3], **Mingqiang Wu**[1,2], **Yuqi Li**[4],
**Xiangqi Li**[1,2], **Zhulin An**[1]*, **Libo Huang**[1], **Yulun Zhang**[5], **Michele Magno**[3], **Yongjun Xu**[1]

[1]State Key Laboratory of AI Safety, Institute of Computing Technology, Chinese Academy of Sciences
[2]University of Chinese Academy of Sciences   [3]ETH Zürich
[4]City College of New York, City University of New York, USA   [5]Shanghai Jiao Tong University
{fengweilun24s,yangchuanguang,lixiangqi24s,anzhulin,xyj}@ict.ac.cn
{haotong.qin,michele.magno}@pbl.ee.ethz.ch, wumingqiang25@mails.ucas.ac.cn,
{yuqili010602,www.huanglibo,yulun100}@gmail.com

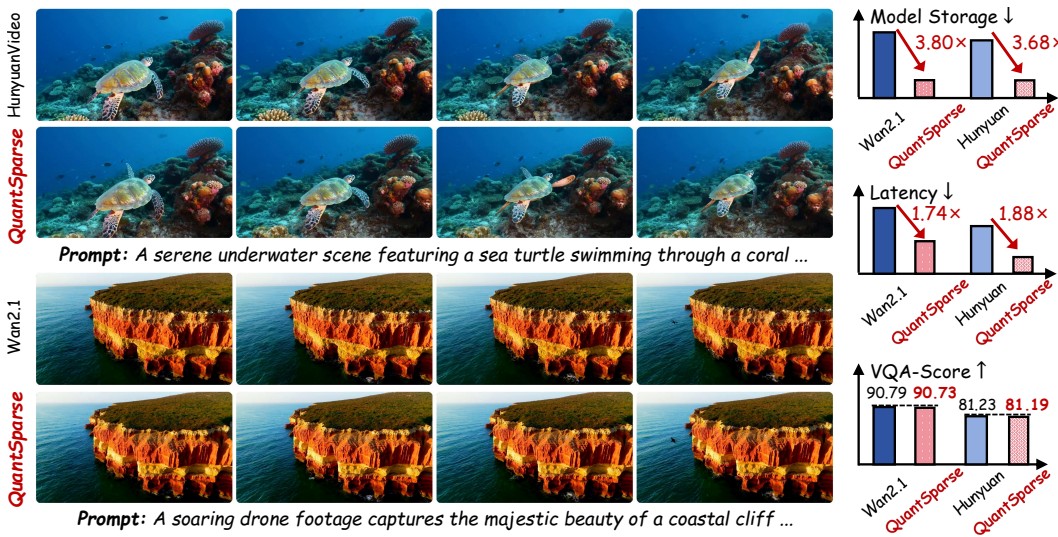

Figure 1: **QuantSparse** effectively quantizes Wan2.1-14B (Wan et al., 2025) and Hunyuan-Video (Kong et al., 2024) to W4A8 with 15% attention density without compromising visual quality.

## ABSTRACT

Diffusion transformers exhibit remarkable video generation capability, yet their prohibitive computational and memory costs hinder practical deployment. Model quantization and attention sparsification are two promising directions for compression, but each alone suffers severe performance degradation under aggressive compression. Combining them promises compounded efficiency gains, but naive integration is ineffective. The sparsity-induced information loss exacerbates quantization noise, leading to amplified attention shifts. To address this, we propose **QuantSparse**, a unified framework that integrates model quantization with attention sparsification. Specifically, we introduce *Multi-Scale Salient Attention Distillation*, which leverages both global structural guidance and local salient supervision to mitigate quantization-induced bias. In addition, we develop *Second-Order Sparse Attention Reparameterization*, which exploits the temporal stability of second-order residuals to efficiently recover information lost under sparsity. Experiments on HunyuanVideo-13B demonstrate that QuantSparse

---

*Corresponding authors: Chuanguang Yang, yangchuanguang@ict.ac.cn; Zhulin An, anzhulin@ict.ac.cn

achieves 20.88 PSNR, substantially outperforming the state-of-the-art quantization baseline Q-VDiT (16.85 PSNR), while simultaneously delivering a **3.68×** reduction in storage and **1.88×** acceleration in end-to-end inference. Our code will be released in https://github.com/wlfeng0509/QuantSparse.

# 1 INTRODUCTION

Recently, Diffusion Transformer (DiT) (Peebles & Xie, 2023) has attracted significant attention due to its outstanding capability in visual generation, particularly in video generation (Liu et al., 2024c; Sun et al., 2024a; HPC-AI, 2024; An et al., 2025). Despite the remarkable progress, state-of-the-art models such as Wan2.1-14B (Wan et al., 2025) still demand extraordinary computational resources: generating a single high-resolution video clip can consume more than **20GB** of GPU memory and take nearly **one hour** of inference time. Such prohibitive memory and latency requirements fundamentally limit the deployment of diffusion-based video generation models in real-world applications, especially under resource-constrained scenarios.

Model quantization (Jacob et al., 2018; Gholami et al., 2022; Ma et al., 2023a;b) and attention sparsification (Xi et al., 2025; Yuan et al., 2024) have emerged as two promising directions for compression and acceleration. Quantization reduces memory footprint and computation by representing weights and activations in compact integer formats, while attention sparsification prunes redundant computations by removing negligible attention scores. However, pushing either technique to the extreme inevitably causes severe degradation. For instance, binary quantization (Zheng et al., 2024b;a) collapses representational capacity, while aggressive sparsification (Xi et al., 2025; Zhang et al., 2025d) discards crucial context information.

Since quantization and sparsification are fundamentally orthogonal, a natural idea is to combine them for compounded efficiency gains while maintaining complementary benefits. Ideally, such integration could approach a Pareto frontier between performance and efficiency. Yet, our empirical analysis shows that **naïvely combining quantization and sparsification leads to severe performance degradation**. We attribute this to an *amplified attention shift*: while sparsification removes low-magnitude attention weights, quantization introduces systematic perturbations to the remaining attention products. These two effects reinforce each other, producing compounded distortions in attention distributions and severely impairing fine-grained dependency modeling in video generation.

To overcome this challenge, we propose **QuantSparse**, a unified compression framework that synergistically integrates model quantization and attention sparsification as shown in Fig. 2. QuantSparse introduces two novel techniques. First, *Multi-Scale Salient Attention Distillation (MSAD)*. We design a memory-efficient distillation scheme that balances global and local supervision. Specifically, we employ *global guidance* by distilling attention patterns on downsampled token sequences to capture coarse structural topology, while *local guidance* focuses high-resolution supervision on a small set of salient tokens that dominate the attention distribution. Second, *Second-Order Sparse Attention Reparameterization (SSAR)*. We exploit the temporal stability of *second-order residuals* to recover information lost due to sparsity. Furthermore, we introduce singular value decomposition (SVD) projection onto dominant principal components, enabling a lightweight yet accurate correction mechanism that restores fine-grained attention outputs at negligible computational overhead.

Our contributions can be summarized as follows:

1. We provide formal analysis of the *amplified attention shift* problem, showing that naive integration of quantization and sparsification severely damages video generation quality.

2. We propose **QuantSparse**, a unified compression framework that seamlessly combines model quantization and attention sparsification, breaking the traditional trade-off between efficiency and performance.

3. We introduce two key techniques: *Multi-Scale Salient Attention Distillation* for robust attention alignment and *Second-Order Sparse Attention Reparameterization* for temporally stable correction for efficient yet accurate approximation of full-attention outputs.

4. Extensive experiments on large-scale video generation models ranging from 1.3B to 14B parameters demonstrate that QuantSparse achieves superior efficiency–quality trade-offs,

Figure 2: **Overview of proposed QuantSparse. Left:** During calibration, we apply two parallel attention distillation branch for efficient and robust attention alignment. **Right:** During inference, we apply an accurate attention approximation using temporal stable second-order residual.

> outperforming both quantization-only and sparsification-only baselines, while preserving state-of-the-art performance.

## 2 RELATED WORKS

### 2.1 SPARSE ATTENTION IN DIFFUSION MODELS

Sparse attention has been extensively explored in transformer-based models to accelerate attention computation (Lu et al., 2025; Yuan et al., 2025; Lou et al., 2024; Gao et al., 2024; Zhang et al., 2025b). In large language models, common designs include sliding-window (Xiao et al., 2024a;b; Zhang et al., 2023) and sink-based patterns (Fu et al.; Xiao et al., 2023b). For diffusion-based visual generation, spatial window masks (Yuan et al., 2024; Zhang et al., 2025c; Ren et al., 2025) and spatial-temporal masks (Xi et al., 2025) have been proposed. Other approaches dynamically generate masks via sampling (Zhang et al., 2025b) or low-resolution attention (Zhang et al., 2025d), though at higher computational cost. However, these works mainly focus on preserving the original attention pattern, while the adaptation to other acceleration techniques that alter attention distributions, such as quantization, remains underexplored.

### 2.2 QUANTIZATION IN DIFFUSION MODELS

Quantization (Gholami et al., 2022; Chitty-Venkata et al., 2023; Pilipović et al., 2018; Ma et al., 2024b;c; Feng et al., 2025b; Ding et al., 2024; Feng et al., 2025d) reduces model precision to improve efficiency and has been applied to diffusion-based visual generation (Shang et al., 2023; Li et al., 2024b; He et al., 2024; Huang et al., 2024a; He et al., 2023; Feng et al., 2025a; Wu et al., 2024; Zheng et al., 2024a;b; Li et al., 2024a). For video generation, some works target the attention module (Zhang et al., 2024b;a; 2025a), but often keep linear operations in high precision. Other methods focus on quantizing linear layers: Q-DiT (Chen et al., 2024) uses automatic granularity allocation; ViDiT-Q (Zhao et al., 2024) adopts a static–dynamic strategy; Q-VDiT (Feng et al., 2025c) introduces temporal distillation. These methods primarily pursue acceleration via quantization, without exploring its synergy with sparse attention. In this work, we integrate the two orthogonal compression techniques to enhance the efficiency and practicality of video generation models.

## 3 METHODS

### 3.1 PRELIMINARY

#### 3.1.1 POST-TRAINING QUANTIZATION (PTQ)

Model Quantization (Gholami et al., 2022; Chitty-Venkata et al., 2023) reduces weights/activations from floating-point (FP32) to low-bit integers (e.g., INT8). Given an floating-point tensor $\mathbf{X} \in \mathbb{R}^d$ with dimension $d$, quantization maps $\mathbf{X}$ to a discrete representation $\mathbf{X_Q} \in \{0, 1, \ldots, 2^b - 1\}^d$ as:

$$\mathbf{X}_Q = \text{clip}\left(\left\lfloor \frac{\mathbf{X}}{s} \right\rceil + z, 0, 2^b - 1\right), \quad Q(\mathbf{X}) = s \cdot (\mathbf{X}_Q - z), \tag{1}$$

with scale $s$, zero-point $z$, and bit-width $b$, $Q(\mathbf{X})$ denotes the de-quantized value. Post-training Quantization (PTQ) (Wei et al., 2024; Wu et al., 2024) calibrates $(s, z)$ on a small dataset by minimizing reconstruction error:

$$\mathcal{L}_{\text{quant}} = \min_{s,z} \sum_{\mathbf{X}_i \in \mathcal{D}_{\text{cal}}} \|\mathbf{X}_i - Q(\mathbf{X}_{Q_i}; s, z)\|_2^2. \tag{2}$$

Notably, PTQ avoids retraining the model weights, thus being computationally efficient.

### 3.1.2 SPARSE ATTENTION

Sparse attention (Zhang et al., 2025b; Xi et al., 2025; Yuan et al., 2024) prunes token pairs via a mask $\mathbf{M} \in \{0,1\}^{L \times L}$, reducing complexity from $\mathcal{O}(L^2)$ to near-linear ($L$ is the sequence length). Given $\mathbf{X} \in \mathbb{R}^{L \times d_{\text{in}}}$ and query, key, value projection matrices $\mathbf{W}_q, \mathbf{W}_k, \mathbf{W}_v \in \mathbb{R}^{d_{\text{out}} \times d_{\text{in}}}$, sparse attention computes:

$$\mathbf{Q} = \mathbf{X}\mathbf{W}_q^\top, \quad \mathbf{K} = \mathbf{X}\mathbf{W}_k^\top, \quad \mathbf{V} = \mathbf{X}\mathbf{W}_v^\top,$$

$$\text{SparseAttention}(\mathbf{Q}, \mathbf{K}, \mathbf{V}; \mathbf{M}) = \text{softmax}\left(\frac{\mathbf{Q}\mathbf{K}^\top}{\sqrt{d_k}} \odot \mathbf{M}\right)\mathbf{V}, \tag{3}$$

where $\odot$ denotes element-wise multiplication.

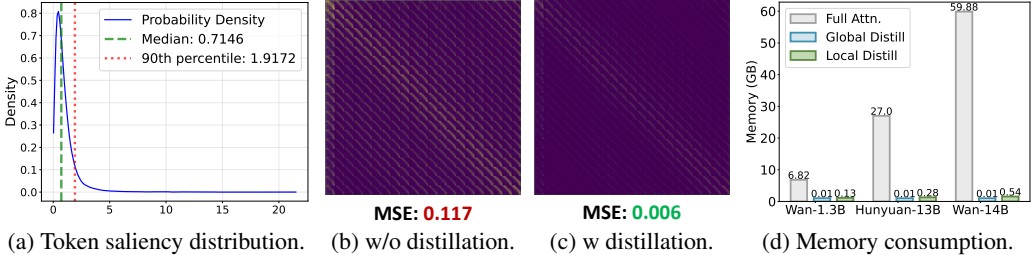

(a) Token saliency distribution.    (b) w/o distillation.    (c) w distillation.    (d) Memory consumption.

Figure 3: **The motivation and effect of Multi-Scale Salient Attention Distillation.** (a): Token saliency distribution of Wan2.1-1.3B (Wan et al., 2025) *block19 head1*. Only less than 10% tokens are salient. (b)(c): Visualization of attention difference between quantized model and FP model. (d): Memory consumption of different attention distillation.

### 3.2 MULTI-SCALE SALIENT ATTENTION DISTILLATION

The combination PTQ and sparse attention offers a promising route toward efficient video generation. However, naively integrating these techniques results in severe performance degradation.

**Proposition 3.1.** *Quantization injects noise $\epsilon$ into the QK dot product $\mathbf{Q}\mathbf{K}^\top$, yielding a systematic bias $\delta$:*

$$\hat{\mathbf{Q}} = Q(\mathbf{X})Q(\mathbf{W}_q)^\top, \ \hat{\mathbf{K}} = Q(\mathbf{X})Q(\mathbf{W}_k)^\top,$$

$$\hat{\mathbf{Q}}\hat{\mathbf{K}}^\top = \mathbf{Q}\mathbf{K}^\top + \epsilon, \quad where \quad \|\epsilon\|_F \leq \delta. \tag{4}$$

*The parallel error caused by quantization and sparse attention further leads to a compounded shift:*

$$\Delta_{\text{total}} = \Delta_{\text{sparse}} + \Delta_{\text{quant}} + \mathcal{O}(\|\epsilon\|_F \cdot \|\mathbf{M}\|_0). \tag{5}$$

Proposition 3.1 indicates that the joint of quantization and sparse attention introduces an *amplified attention shift* (see Fig. 3b), resulting in notable attention degradation. A straightforward mitigation strategy is to perform attention distillation during PTQ. However, for large-scale video generation

models (e.g., with $L > 10^4$ for HunyuanVideo (Kong et al., 2024)), storing the full attention matrices is prohibitively expensive as shown in Fig. 3d, incurring $\mathcal{O}(L^2)$ memory and compute overhead.

To address this, we propose *Multi-Scale Salient Attention Distillation* (MSAD), a memory-efficient framework that distills attention across multiple resolutions, preserving both global structure and local saliency without excessive resource consumption. MSAD employs two complementary guidance mechanisms: *global guidance* for high-level structural supervision, and *local guidance* for fine-grained detail preservation.

**Global Guidance.** Our approach exploits the intrinsic *locality* of video data: patially adjacent tokens exhibit high similarity due to temporal smoothness and spatial continuity (Ren et al., 2025; Xi et al., 2025; Yuan et al., 2024; Feng et al., 2024). To efficiently capture global attention patterns, we downsample $\mathbf{Q}$ and $\mathbf{K}$ via average pooling with stride $s$, producing low-resolution features $\tilde{\mathbf{Q}}, \tilde{\mathbf{K}} \in \mathbb{R}^{\tilde{L} \times d_k}$ where $\tilde{L} = L/s^2 \ll L$. The global distillation is computed as:

$$\mathbf{A}_{\text{global}} = \text{softmax}\left(\frac{\tilde{\mathbf{Q}}\tilde{\mathbf{K}}^\top}{\sqrt{d_k}}\right), \ \mathcal{L}_{\text{global}} = \text{MSE}\left(\mathbf{A}_{\text{global}}^{\text{FP}} \| \mathbf{A}_{\text{global}}^{\text{quant}}\right), \tag{6}$$

where MSE denotes the Mean Square Error. This approach requires only $\mathcal{O}(\tilde{L}^2)$ complexity, which is $s^2$ times cheaper than full attention.

**Local Guidance.** While global guidance ensures structural fidelity, it fails to capture the fine-grained details crucial for high-quality video synthesis. We further observe that the attention saliency in video models is highly *skewed*: only a small subset of tokens dominates the attention mass (see Fig 3a). Formally, we define the token saliency as:

$$\mathbf{A} = \text{softmax}(\mathbf{Q}\mathbf{K}^\top/\sqrt{d_k}) \in \mathbb{R}^{h,L,L}, \ s_j = \sum_{h,i} A_{h,i,j}, \tag{7}$$

where $h$ denotes the attention head, $i$ denotes the key token index, and $s_j$ measures the aggregate attention received by token $j$. Empirically, $s_j$ follows a heavy-tailed distribution, with fewer tokens accounting for the majority of attention mass (**we provide more analysis in Appendix Sec. F**). We exploit this by selecting the top-$k$ queries $\mathcal{I} = \{j \mid s_j \text{ is top-}k\}$ from the FP model and computing high-resolution attention *only* for these salient queries:

$$\mathbf{A}_{\text{local}} = \text{softmax}\left(\frac{\mathbf{Q}_{\mathcal{I},:}\mathbf{K}^\top}{\sqrt{d_k}}\right), \ \mathcal{L}_{\text{local}} = \text{MSE}\left(\mathbf{A}_{\text{local}}^{\text{FP}} \| \mathbf{A}_{\text{local}}^{\text{quant}}\right), \tag{8}$$

where $\mathbf{Q}_{\mathcal{I},:} \in \mathbb{R}^{k \times d_k}$. Local distillation focuses supervision on high-impact regions at minimal cost.

**Integration and Optimization.** We combine both guidance terms into a unified distillation object:

$$\mathcal{L}_{\text{distill}} = \mathcal{L}_{\text{quant}} + \lambda_{\text{global}}\mathcal{L}_{\text{global}} + \lambda_{\text{local}}\mathcal{L}_{\text{local}}, \tag{9}$$

where $\lambda_{\text{global}}$ and $\lambda_{\text{local}}$ balance the two guidance component. During PTQ calibration, we optimize the quantization parameters over $\mathcal{D}_{\text{cal}}$ to minimize $\mathcal{L}_{\text{distill}}$, aligning the quantized attention with its FP counterpart. As shown in Fig 3b and Fig 3c, MSAD substantially reduces attention shift, enabling robust integration of quantization and sparse attention in video generation.

### 3.3 SECOND-ORDER SPARSE ATTENTION REPARAMETERIZATION

While the proposed MSAD mitigates the quantization-induced attention shift during calibration phase by aligning attention maps, the intrinsic bottleneck of sparse attention (i.e., the unavoidable discard of low-magnitude yet non-trivial attention connections) still exacerbates the amplified attention shift, especially under high sparsity rates (Xi et al., 2025; Zhang et al., 2025b). We formalize this deviation at denoising timestep $t$ in the diffusion process as: $\Delta^{(t)} = \mathbf{A}_{\text{full}}^{(t)} - \mathbf{A}_{\text{sparse}}^{(t)}$, where $\mathbf{A}_{\text{full}}$ and $\mathbf{A}_{\text{sparse}}$ denote the full-attention and sparse attention. We define this deviation $\Delta^{(t)}$ as the *first-order residual*. This residual is intrinsic to sparsity and cannot be recovered through attention

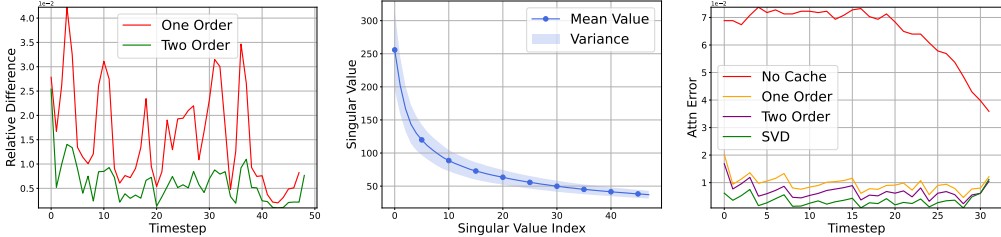

(a) Residual temporal difference.    (b) Singular value distribution of all    (c) Attention error comparison.
timesteps.

Figure 4: **The motivation and effect of Second-Order Sparse Attention Reparameterization.** The results are from HunyuanVideo-13B (Kong et al., 2024) *single_transformer_block.10* under W4A8. **We provide more visualization and analysis in Appendix Sec. G.**

distillation alone. Prior work (Yuan et al., 2024) exploits temporal coherence in video generation by assuming that residuals are invariant across timesteps:

$$\Delta^{(t')} \approx \Delta^{(t)} \quad \forall t, t', \tag{10}$$

Under this assumption, one can cache a reference residual $\Delta^{(t_{\text{ref}})}$ from a chosen timestep and reuse it across the successive timesteps, yielding a *first-order sparse attention reparameterization*:

$$\mathbf{A}_{\text{full}}^{(t)} - \mathbf{A}_{\text{sparse}}^{(t)} \approx \mathbf{A}_{\text{full}}^{(t_{\text{ref}})} - \mathbf{A}_{\text{sparse}}^{(t_{\text{ref}})} = \Delta^{(t_{\text{ref}})} \Rightarrow \hat{\mathbf{A}}^{(t)} = \mathbf{A}_{\text{sparse}}^{(t)} + \underbrace{\Delta^{(t_{\text{ref}})}}_{\text{cached}}, \tag{11}$$

**Proposition 3.2.** *Let $\mathbf{A}_{\text{s, q}}^{(t)}$ denote the quantized sparse attention output. The quantization-induced perturbation $\epsilon^{(t)}$ (as defined in Eq. 4) modifies the one-order residual to:*

$$\Delta_{\text{quant}}^{(t)} = \mathbf{A}_{\text{full}}^{(t)} - \mathbf{A}_{\text{s, q}}^{(t)} = \Delta^{(t)} + \epsilon^{(t)} + \mathcal{O}(\|\epsilon^{(t)}\|_F \cdot \|\mathbf{M}\|_0), \tag{12}$$
$$\Rightarrow \Delta_{\text{quant}}^{(t')} \neq \Delta_{\text{quant}}^{(t)}, \quad \text{for} \quad t' \neq t.$$

Proposition 3.2 indicates that, unlike $\Delta^{(t)}$, $\Delta_{\text{quant}}^{(t)}$ varies with $\epsilon^{(t)}$ due to the quantization noise (Wu et al., 2024; Zhao et al., 2024; He et al., 2023) which violating Eq. 10. We visualize this variance of $\Delta^{(t)} - \Delta^{(t-1)}$ in Fig. 4a. This temporal variance undermines the accuracy of Eq. 11, causing non-negligible attention errors when *first-order reparameterization* is applied after quantization.

**Proposition 3.3.** *Although $\Delta_{\text{quant}}^{(t)}$ is unstable, we observe that the second-order residual $\hat{\Delta}_{\text{quant}}^{(t)} := \Delta_{\text{quant}}^{(t)} - \Delta_{\text{quant}}^{(t-1)}$ exhibits significantly higher temporal stability:*

$$\mathbb{E}_t \left[ \left\| \hat{\Delta}_{\text{quant}}^{(t)} - \hat{\Delta}_{\text{quant}}^{(t')} \right\|_F \right] \leq \mathbb{E}_t \left[ \left\| \Delta_{\text{quant}}^{(t)} - \Delta_{\text{quant}}^{(t')} \right\|_F \right] \quad \text{for} \quad |t - t'| \leq \tau. \tag{13}$$

We visualize the empirical analysis results in Fig. 4a. This stability arises because quantization noise $\epsilon^{(t)}$ follows a *slow-varying stochastic process* in diffusion process (Ma et al., 2024a; Liu et al., 2024a): adjacent timesteps share similar distributions, rendering $\epsilon^{(t)} - \epsilon^{(t-1)}$ approximately stationary. Leveraging this property, we propose *second-order sparse attention reparameterization*:

$$(\mathbf{A}_{\text{full}}^{(t)} - \mathbf{A}_{\text{s,q}}^{(t)}) - (\mathbf{A}_{\text{full}}^{(t_{\text{ref}})} - \mathbf{A}_{\text{s,q}}^{(t_{\text{ref}})}) \approx (\mathbf{A}_{\text{full}}^{(t_{\text{ref}})} - \mathbf{A}_{\text{s,q}}^{(t_{\text{ref}})}) - (\mathbf{A}_{\text{full}}^{(t'_{\text{ref}})} - \mathbf{A}_{\text{s,q}}^{(t'_{\text{ref}})}) = \hat{\Delta}_{\text{quant}}^{(t_{\text{ref}})},$$
$$\Rightarrow \tilde{\mathbf{A}}^{(t)} = \mathbf{A}_{\text{s,q}}^{(t)} + (\mathbf{A}_{\text{full}}^{(t_{\text{ref}})} - \mathbf{A}_{\text{s,q}}^{(t_{\text{ref}})}) + \hat{\Delta}_{\text{quant}}^{(t_{\text{ref}})},$$
$$= \mathbf{A}_{\text{s,q}}^{(t)} + \underbrace{\Delta_{\text{quant}}^{(t_{\text{ref}})} + \hat{\Delta}_{\text{quant}}^{(t_{\text{ref}})}}_{\text{cached}}. \tag{14}$$

Table 1: Text-to-Video generation results on Wan2.1-1.3B. Density is the attention density. Full Prec. denotes Full Precision model. **Bold**: the best result.

| Method | #Bits (W/A) | Density↓ | Quality | | | | | |
| --- | --- | --- | --- | --- | --- | --- | --- | --- |
| | | | *Video Quality Metrics* | | | *FP Diff. Metrics* | | |
| | | | CLIPSIM↑ | VQA↑ | ΔFSCore↓ | PSNR↑ | SSIM↑ | LPIPS↓ |
| Wan2.1 1.3B (CFG = 6.0, $480 \times 832p$, frames = 80) | | | | | | | | |
| Full Prec. | 16/16 | 100% | 0.191 | 73.12 | 0.000 | - | - | - |
| PTQ4DiT | 6/6 | 100% | 0.182 | 36.79 | 2.287 | 10.20 | 0.343 | 0.598 |
| Q-DiT | 6/6 | 100% | 0.183 | 39.21 | 2.125 | 10.36 | 0.351 | 0.577 |
| SmoothQuant | 6/6 | 100% | 0.184 | 40.57 | 2.008 | 10.44 | 0.353 | 0.574 |
| QuaRot | 6/6 | 100% | 0.190 | 42.81 | 1.754 | 10.71 | 0.379 | 0.571 |
| ViDiT-Q | 6/6 | 100% | 0.190 | 50.85 | 1.253 | 11.02 | 0.385 | 0.526 |
| Q-VDiT | 6/6 | 100% | 0.191 | 75.20 | 0.982 | 12.06 | 0.405 | 0.474 |
| QuaRot+DFT | 6/6 | 40% | 0.183 | 36.79 | 2.297 | 11.29 | 0.321 | 0.546 |
| QuaRot+Jenga | 6/6 | 40% | 0.184 | 38.78 | 2.165 | 11.32 | 0.329 | 0.543 |
| QuaRot+SVG | 6/6 | 40% | 0.183 | 41.93 | 1.940 | 11.43 | 0.331 | 0.541 |
| Q-VDiT+DFT | 6/6 | 40% | 0.188 | 47.33 | 1.377 | 11.06 | 0.345 | 0.577 |
| Q-VDiT+Jenga | 6/6 | 40% | 0.189 | 53.52 | 1.087 | 11.21 | 0.345 | 0.583 |
| Q-VDiT+SVG | 6/6 | 40% | 0.191 | 55.92 | 0.942 | 11.61 | 0.384 | 0.508 |
| **QuantSparse** | 6/6 | 40% | **0.193** | **78.35** | **0.055** | **15.51** | **0.511** | **0.324** |
| PTQ4DiT | 4/8 | 100% | 0.181 | 30.26 | 2.574 | 10.00 | 0.318 | 0.603 |
| Q-DiT | 4/8 | 100% | 0.182 | 32.57 | 2.767 | 10.11 | 0.320 | 0.594 |
| SmoothQuant | 4/8 | 100% | 0.182 | 34.82 | 2.174 | 10.20 | 0.327 | 0.569 |
| QuaRot | 4/8 | 100% | 0.185 | 65.15 | 1.870 | 11.72 | 0.349 | 0.514 |
| ViDiT-Q | 4/8 | 100% | 0.186 | 63.21 | 1.698 | 11.24 | 0.351 | 0.526 |
| Q-VDiT | 4/8 | 100% | 0.190 | 56.45 | 2.240 | 11.01 | 0.394 | 0.565 |
| QuaRot+DFT | 4/8 | 40% | 0.187 | 32.23 | 2.329 | 10.32 | 0.360 | 0.583 |
| QuaRot+Jenga | 4/8 | 40% | 0.191 | 32.83 | 2.148 | 10.33 | 0.346 | 0.578 |
| QuaRot+SVG | 4/8 | 40% | 0.190 | 32.48 | 2.088 | 10.58 | 0.370 | 0.576 |
| Q-VDiT+DFT | 4/8 | 40% | 0.185 | 45.60 | 2.907 | 10.03 | 0.331 | 0.594 |
| Q-VDiT+Jenga | 4/8 | 40% | 0.185 | 47.61 | 3.000 | 10.04 | 0.334 | 0.596 |
| Q-VDiT+SVG | 4/8 | 40% | 0.184 | 51.84 | 3.035 | 10.07 | 0.342 | 0.592 |
| **QuantSparse** | 4/8 | 40% | **0.193** | **81.09** | **0.576** | **15.22** | **0.502** | **0.338** |

**Theorem 3.4.** *When Proposition 3.3 holds, the expected approximation error of sparse attention satisfies:*

$$\mathbb{E}_t \underbrace{\left[ \left\| \mathbf{A}_{\text{full}}^{(t)} - \tilde{\mathbf{A}}_{\text{s,q}}^{(t)} \right\|_F \right]}_{\text{second-order}} \leq \mathbb{E}_t \underbrace{\left[ \left\| \mathbf{A}_{\text{full}}^{(t)} - \hat{\mathbf{A}}_{\text{s,q}}^{(t)} \right\|_F \right]}_{\text{first-order}} \quad \text{for} \quad |t - t'| \leq \tau. \tag{15}$$

Theorem 3.4 indicates *two-order* guaranteeing tighter full-attention approximation than the first-order method. Also $\Delta_{\text{quant}}^{(t_{\text{ref}})} + \hat{\Delta}_{\text{quant}}^{t_{\text{ref}}}$ in Eq. 14 can be jointly cached, without any additional storage burden compared with one-order residual. We further reduce the temporal variance of $\hat{\Delta}_{\text{quant}}$ by projecting it onto its most stable subspace. Empirically, the top-$r$ principal components from the singular value decomposition (SVD) of $\hat{\Delta}_{\text{quant}}$ capture the dominant, temporally stable patterns (see Fig. 4b). Critically, the dominant principal component exhibit exceptional temporal stability, which inspired us to project residuals onto the top-$r$ extracted stable components:

$$\text{SVD}(\hat{\Delta}_{\text{quant}}) = \mathbf{S}\mathbf{U}\mathbf{V}^\top, \ \tilde{\Delta}_{\text{quant}} := \mathbf{S}_{:,:r}\mathbf{U}_{:r,:r}\mathbf{V}_{:,:r}^\top,$$
$$\tilde{\mathbf{A}}^{(t)} = \mathbf{A}_{\text{s,q}}^{(t)} + \underbrace{\Delta_{\text{quant}}^{(t_{\text{ref}})} + \tilde{\Delta}_{\text{quant}}^{t_{\text{ref}}}}_{\text{cached}}. \tag{16}$$

We apply the sparse attention for inference with a fixed cache-refreshing interval (5 in experiments) for full-attention calculation. As visualized in Fig. 4c, SVD-based second-order reparameterization further suppresses temporal variance, yielding accurate full-attention approximation results.

Table 2: Video generation on large video generation models. **Bold**: the best result. Underline: the second best result.

| Method | #Bits (W/A) | Density↓ | Quality | | | | | | Latency & Speed | |
|---|---|---|---|---|---|---|---|---|---|---|
| | | | Video Quality Metrics | | | FP Diff. Metrics | | | DiT Time↓ | Speedup↑ |
| | | | CLIPSIM↑ | VQA↑ | ΔFSCore↓ | PSNR↑ | SSIM↑ | LPIPS↓ | | |
| HunyuanVideo 13B (CFG = 6.0, $720 \times 1280p$, frames = 60) | | | | | | | | | | |
| Full Prec. | 16/16 | 100% | 0.184 | 81.23 | 0.000 | - | - | - | 1264s | 1.00× |
| SmoothQuant | 4/8 | 100% | 0.178 | 42.21 | 1.194 | 15.44 | 0.479 | 0.583 | 1149s | 1.10× |
| QuaRot | 4/8 | 100% | 0.180 | 42.89 | 0.708 | 15.46 | 0.502 | 0.528 | 1149s | 1.10× |
| ViDiT-Q | 4/8 | 100% | 0.181 | 49.82 | 1.254 | 15.75 | 0.534 | 0.489 | 1149s | 1.10× |
| Q-VDiT | 4/8 | 100% | 0.182 | 67.95 | 1.168 | 16.85 | 0.605 | 0.461 | 1155s | 1.09× |
| QuaRot+SVG | 4/8 | 25% | 0.181 | 43.34 | 0.900 | 15.39 | 0.497 | 0.530 | 731s | 1.73× |
| Q-VDiT+SVG | 4/8 | 25% | 0.182 | 70.99 | 1.379 | 16.71 | 0.595 | 0.458 | 743s | 1.70× |
| QuaRot+SVG | 4/8 | 15% | 0.181 | 41.40 | 1.004 | 15.34 | 0.494 | 0.536 | 671s | 1.88× |
| Q-VDiT+SVG | 4/8 | 15% | 0.182 | 76.30 | 1.393 | 16.66 | 0.591 | 0.460 | 687s | 1.84× |
| **QuantSparse** | 4/8 | 25% | 0.183 | 79.05 | **0.014** | 20.86 | 0.675 | **0.272** | 731s | 1.73× |
| **QuantSparse** | 4/8 | 15% | **0.184** | **81.19** | 0.016 | **20.88** | **0.678** | 0.273 | 671s | 1.88× |
| Wan2.1 14B (CFG = 5.0, $720 \times 1280p$, frames = 80) | | | | | | | | | | |
| Full Prec. | 16/16 | 100% | 0.182 | 90.79 | 0.000 | - | - | - | 4031s | 1.00× |
| SmoothQuant | 4/8 | 100% | 0.180 | 73.11 | 0.875 | 13.70 | 0.423 | 0.510 | 3425s | 1.18× |
| QuaRot | 4/8 | 100% | 0.182 | 85.91 | 0.753 | 13.79 | 0.431 | 0.494 | 3425s | 1.18× |
| ViDiT-Q | 4/8 | 100% | 0.182 | 83.13 | 0.496 | 15.12 | 0.487 | 0.425 | 3425s | 1.18× |
| Q-VDiT | 4/8 | 100% | 0.182 | 83.76 | 0.343 | 15.85 | 0.512 | 0.398 | 3457s | 1.17× |
| QuaRot+SVG | 4/8 | 25% | 0.182 | 85.66 | 0.134 | 13.70 | 0.427 | 0.487 | 2594s | 1.55× |
| Q-VDiT+SVG | 4/8 | 25% | 0.182 | 87.89 | 0.310 | 15.48 | 0.507 | 0.409 | 2635s | 1.53× |
| QuaRot+SVG | 4/8 | 15% | 0.182 | 81.93 | 0.152 | 13.40 | 0.415 | 0.494 | 2315s | 1.74× |
| Q-VDiT+SVG | 4/8 | 15% | 0.181 | 82.31 | 0.411 | 15.18 | 0.493 | 0.429 | 2372s | 1.70× |
| **QuantSparse** | 4/8 | 25% | **0.183** | **91.98** | 0.056 | **18.72** | **0.630** | **0.240** | 2594s | 1.55× |
| **QuantSparse** | 4/8 | 15% | 0.182 | 90.73 | **0.042** | 18.22 | 0.605 | 0.272 | 2315s | 1.74× |

## 3.4 OVERALL PIPELINE

Our proposed QuantSparse framework consists of two component as shown in Fig. 2: *MSAD* for attention distillation during calibration and *SSAR* for dynamic attention reparameterization during inference. The detailed overall pipeline is provided in Appendix Algorithm 1.

## 4 EXPERIMENTS

### 4.1 EXPERIMENTAL AND EVALUATION SETTINGS

**Evaluation Settings.** We apply QuantSparse to HunyuanVideo-13B (Kong et al., 2024), Wan2.1-1.3B and 14B (Wan et al., 2025) with 50 sampling steps. We employ two types of metrics: (1) Multi-aspects metrics evaluation: including CLIPSIM (Wu et al., 2021), VQA (Wu et al., 2023), FlowScore (Liu et al., 2024b), PSNR, SSIM, and LPIPS (Zhang et al., 2018). All metrics are evaluated on the prompt sets used in (Zhao et al., 2024; Feng et al., 2025c) (2) Benchmark evaluation: We select 8 major dimensions from Vbench (Huang et al., 2024b) following prior works (Zhao et al., 2024; Chen et al., 2024; Feng et al., 2025c). For bit setting, we use W6A6 and W4A8 following prior work (Zhao et al., 2024; Chen et al., 2024; Wu et al., 2024), since they can bring more compression effects and ensure the performance.

**Baseline Methods.** We select PTQ4DiT (Wu et al., 2024), Q-DiT (Chen et al., 2024), ViDiT-Q (Zhao et al., 2024), and Q-VDiT (Feng et al., 2025c) for diffusion baseline. We also compare with strong LLM baseline SmoothQuant (Xiao et al., 2023a) and QuaRot (Ashkboos et al., 2024). For sparsification, we compare with DiTFastAttn (DFT) (Yuan et al., 2024) (cache-based), Jenga (Zhang et al., 2025d) (dynamic pattern), and SparseVideoGen (SVG) (Xi et al., 2025) (static pattern).

**Implementation Detail.** Same with prior works (Zhao et al., 2024; Ashkboos et al., 2024; Feng et al., 2025c), we adopt channel-wise weight quantization and dynamic token-wise activation quantization. We follow block-wise post-training strategy used in (Wu et al., 2024; Chen et al., 2024; Sun et al., 2024b) for calibration. **More details can be found in Appendix C.**

### 4.2 MAIN RESULTS

We present multi-aspects metrics evaluation results on HunyuanVideo (Kong et al., 2024) and Wan2.1-14B (Wan et al., 2025) in Tab. 2. It can be seen that the existing SOTA quantization methods have a significant performance degradation after applying sparse attention. But QuantSparse still maintains high generation performance even at high sparsity. It is worth mentioning that QuantSparse even surpasses the existing quantization-only methods under the low-bit settings of W6A6 and W4A8.

Table 3: Ablation results of each component.

| Method | VQA↑ | PSNR↑ | SSIM↑ | LPIPS↓ |
|---|---|---|---|---|
| Distillation Analysis | | | | |
| None | 81.92 | 14.35 | 0.486 | 0.425 |
| *Global* | 85.26 | 16.01 | 0.547 | 0.349 |
| *Local* | 86.95 | 16.82 | 0.561 | 0.325 |
| **MSAD** | **91.98**$_{+10.06}$ | **18.72**$_{+4.37}$ | **0.630**$_{+0.144}$ | **0.240**$_{-0.185}$ |
| Cache Analysis | | | | |
| None | 68.00 | 14.16 | 0.470 | 0.445 |
| *First* | 70.82 | 17.08 | 0.572 | 0.285 |
| *Second* | 89.73 | 18.68 | 0.616 | 0.258 |
| **SSAR** | **91.98**$_{+23.98}$ | **18.72**$_{+4.56}$ | **0.630**$_{+0.160}$ | **0.240**$_{-0.205}$ |

Compared with the Full-Precision (FP) model, QuantSparse even maintains almost lossless performance. For example, for HunyuanVideo under W6A6, QuantSparse achieved a VQA score of 82.26 with only 15% attention density, far exceeding current SOTA method Q-VDiT (Feng et al., 2025c) of 73.68, and even surpassing the FP model of 81.23. **We present more baseline methods comparison in Appendix Sec. D, and comprehensive VBench evaluation results in Appendix Sec. E.** We also observed that QuantSparse slightly outperforms Full Precision model on certain metrics. This slight outperformance of QuantSparse can be attributed to its focus on task-critical tokens and reduced attention to noisy or irrelevant tokens, as shown in our saliency analysis. Additionally, the SSAR module stabilizes sparse attention, reducing quantization noise and improving temporal consistency. These effects, combined with targeted compression, allow QuantSparse to maintain near-lossless quality while offering substantial compression and acceleration. We also visualized the generated videos in Fig. 5. Compared with FP model, QuantSparse achieves almost lossless generation performance while other methods have notable quality degradation. **We provide more visual comparison results in Appendix Sec. M.**

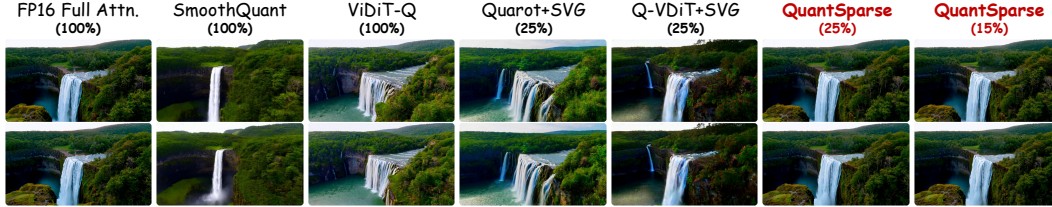

Figure 5: Visual comparison on Wan2.1-14B under W4A8 quantization setting. We uniformly sample two frames for visualization. '(xx%)' denotes the attention density.

## 4.3 ABLATION STUDY

We conduct ablation study on proposed Multi-Scale Salient Attention Distillation (MSAD) and Second-Order Sparse Attention Reparameterization (SSAR) on Wan2.1-14B under W4A8 in Tab. 3.

**Effect of attention distillation.** Compared with no distillation, both proposed attention guidance can enhance the model performance. The combined MSAD further improves PSNR from 14.35 to 18.72, demonstrating the effect of attention distillation.

**Effect of attention reparameterization.** Compared with naive sparse attention, first-order residual can reduce the attention error, demonstrating the effectiveness of attention reparameterization. Our proposed SSAR achieves the best approximation performance by reducing both the quantization-induced error and temporal variance.

**Effect of cache-interval.** We also supplement the ablation and the results are shown in Tab. 5. While shorter intervals yield higher PSNR and SSIM, indicating better performance, they also result in a reduced speedup ($1.65\times$ and $1.69\times$ respectively). For instance, interval=3 achieves the highest PSNR (18.86) but sacrifices a noticeable amount of the potential speedup (9%). Longer intervals increasing the interval to 6 provides a slightly higher speedup ($1.76\times$). However, this comes at the cost of a degradation in performance (PSNR drops to 17.72). We choose interval=5 is based on its optimal balance between model performance and inference speedup. But we highlight that this

Table 4: Detailed efficiency comparison.

| Method | #Bits (W/A) | Density↓ | Model Overload | | Latency & Speed | |
|---|---|---|---|---|---|---|
| | | | Model Storage↓ | Memory Consumption↓ | DiT Time↓ | Speedup↑ |
| HunyuanVideo 13B (CFG = 6.0, 720 × 1280p, frames = 60) | | | | | | |
| Full Prec. | 16/16 | 100% | 23.88GB | 35.79GB | 1264s | 1.00× |
| QuaRot | 4/8 | 100% | 6.49GB | 24.34GB | 1149s | 1.10× |
| Q-VDiT | 4/8 | 100% | 6.50GB | 24.89GB | 1155s | 1.09× |
| DFT | 16/16 | 25% | 23.88GB | 40.11GB | 792s | 1.60× |
| Jenga | 16/16 | 25% | 23.88GB | 36.92GB | 846s | 1.49× |
| SVG | 16/16 | 25% | 23.88GB | 40.10GB | 786s | 1.61× |
| SVG | 16/16 | 15% | 23.88GB | 40.10GB | 707s | 1.79× |
| **QuantSparse** | 4/8 | 25% | 6.49GB$_{↓3.68×}$ | 27.02GB$_{↓1.32×}$ | 731s | 1.73× |
| **QuantSparse** | 4/8 | 15% | 6.49GB$_{↓3.68×}$ | 27.02GB$_{↓1.32×}$ | 671s | 1.88× |
| Wan2.1 14B (CFG = 5.0, 720 × 1280p, frames = 80) | | | | | | |
| Full Prec. | 16/16 | 100% | 26.61GB | 42.48GB | 4031s | 1.00× |
| QuaRot | 4/8 | 100% | 7.00GB | 26.04GB | 3425s | 1.18× |
| Q-VDiT | 4/8 | 100% | 7.02GB | 26.73GB | 3457s | 1.17× |
| DFT | 16/16 | 25% | 26.61GB | 44.86GB | 3015s | 1.34× |
| Jenga | 16/16 | 25% | 26.61GB | 42.62GB | 3087s | 1.31× |
| SVG | 16/16 | 25% | 26.61GB | 44.07GB | 2987s | 1.35× |
| SVG | 16/16 | 15% | 26.61GB | 44.07GB | 2661s | 1.51× |
| **QuantSparse** | 4/8 | 25% | 7.00GB$_{↓3.80×}$ | 28.14GB$_{↓1.51×}$ | 2594s | 1.55× |
| **QuantSparse** | 4/8 | 15% | 7.00GB$_{↓3.80×}$ | 28.14GB$_{↓1.51×}$ | 2315s | 1.74× |

is a trade-off based on computational resource and all interval settings offer reasonable results and notable acceleration.

**More ablation study about pooling stride $s$, salient token $k$, weight factor $\lambda$, and SVD rank $r$ in Eq. 9 and Eq. 16 in Appendix Sec. H.**

## 4.4 EFFICIENCY ANALYSIS

We present the deployment efficiency in Tab. 4. All the experiments are conducted on a single NVIDIA A800 80G GPU with CUDA 12.4. We use CUTLASS (Thakkar et al., 2023) on top of PyTorch for performing INT matrix multiplication. Existing quantization methods can bring higher model compression, but the effect of inference acceleration is limited. Sparse attention brings significant acceleration, but has almost no model compression, and even brings more memory consumption. QuantSparse combines the advantages of both quantization and sparse attention, bringing significant model compression and acceleration. For Wan2.1-14B (Wan et al., 2025), QuantSparse (15% density) brings $3.80×$ storage compression, $1.51×$ memory saving, and $1.74×$ end-to-end acceleration. **We further report the calibration resource consumption in Appendix Sec. I and report the performance combined with other acceleration methods in Appendix Sec. J.**

Table 5: Ablation study of cache-fresh interval and attention density on W4A8 Wan2.1-14B.

| - | PSNR↑ | SSIM↑ | LPIPS↓ | Speedup↑ |
|---|---|---|---|---|
| Interval Analysis | | | | |
| Interval=3 | 18.86 | 0.631 | 0.243 | 1.65× |
| Interval=4 | 18.48 | 0.617 | 0.260 | 1.69× |
| Interval=5 | 18.22 | 0.605 | 0.272 | 1.74× |
| Interval=6 | 17.72 | 0.566 | 0.321 | 1.76× |
| Density Analysis | | | | |
| Density=25% | 18.72 | 0.630 | 0.240 | 1.55× |
| Density=20% | 18.45 | 0.622 | 0.252 | 1.63× |
| Density=15% | 18.22 | 0.605 | 0.272 | 1.74× |
| Density=10% | 17.73 | 0.589 | 0.288 | 1.80× |

## 5 CONCLUSION

In this paper, we propose QuantSparse, a unified compression framework that effectively combines model quantization and sparse attention. To address the amplified attention shift, we propose Multi-Scale Salient Attention Distillation to efficiently align the attention shift. To address the intrinsic sparsity loss, we propose Second-Order Sparse Attention Reparameterization to utilize decomposed second-order residual for attention approximation. Extensive experiments shown that QuantSparse achieves lossless performance while bringing significant model compression and acceleration.

## 6 ACKNOWLEDGEMENTS

This work is supported by the National Natural Science Foundation of China under Grant Number 62406312 and 62476264, the Beijing Natural Science Foundation under Grant Number 4244098, the Science Foundation of the Chinese Academy of Sciences, and the Swiss National Science Foundation (SNSF) project 200021E_219943 Neuromorphic Attention Models for Event Data (NAMED).

## 7 ETHICS STATEMENT

This research strictly adheres to the ICLR Code of Ethics with no ethics-related risks: it uses public open-source video-generation models (Wan2.1 (Wan et al., 2025) and HunyuanVideo (Kong et al., 2024)) and focuses on algorithmic innovation for inference acceleration and compression, without involving scenarios endangering public safety, infringing privacy, or producing discrimination.

## 8 REPRODUCIBILITY STATEMENT

To ensure reproducibility, experimental configurations, method details, and evaluation metrics are thoroughly described in Sec. 4.1 and Appendix Sec. C. Experimental results of comparative methods are sourced from public literature, and our experiments strictly follow the same configurations as baseline methods for fair comparison. The key codes and the presented video source files are also attached in the supplementary materials. For the theorem used in the paper, we also provided a detailed proof in Appendix Sec. A.

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

## A    PROOF OF THEOREM 3.4.

*Proof of Theorem 3.4.*

For $\tilde{\mathbf{A}}_{\text{s,q}}^{(t)}$, we have:

$$(\mathbf{A}_{\text{full}}^{(t)} - \mathbf{A}_{\text{s,q}}^{(t)}) - (\mathbf{A}_{\text{full}}^{(t_{\text{ref}})} - \mathbf{A}_{\text{s,q}}^{(t_{\text{ref}})}) = \hat{\Delta}_{\text{quant}}^{t}$$
$$\Rightarrow \mathbf{A}_{\text{full}}^{(t)} = \mathbf{A}_{\text{s,q}}^{(t)} + (\mathbf{A}_{\text{full}}^{(t_{\text{ref}})} - \mathbf{A}_{\text{s,q}}^{(t_{\text{ref}})}) + \hat{\Delta}_{\text{quant}}^{t} \quad (17)$$
$$= \mathbf{A}_{\text{s,q}}^{(t)} + \Delta_{\text{quant}}^{(t_{\text{ref}})} + \hat{\Delta}_{\text{quant}}^{t}.$$

Given this, we further have:

$$\mathbf{A}_{\text{full}}^{(t)} - \tilde{\mathbf{A}}_{\text{s,q}}^{(t)} = (\mathbf{A}_{\text{s,q}}^{(t)} + \Delta_{\text{quant}}^{(t_{\text{ref}})} + \hat{\Delta}_{\text{quant}}^{(t)}) - \tilde{\mathbf{A}}_{\text{s,q}}^{(t)}$$
$$= (\mathbf{A}_{\text{s,q}}^{(t)} + \Delta_{\text{quant}}^{(t_{\text{ref}})} + \hat{\Delta}_{\text{quant}}^{(t)}) - (\mathbf{A}_{\text{s,q}}^{(t)} + \Delta_{\text{quant}}^{(t_{\text{ref}})} + \hat{\Delta}_{\text{quant}}^{(t_{\text{ref}})}) \quad (18)$$
$$= \hat{\Delta}_{\text{quant}}^{(t)} - \hat{\Delta}_{\text{quant}}^{(t_{\text{ref}})}.$$

Similarly, for $\hat{\mathbf{A}}_{\text{s,q}}^{(t)}$, we also have:

$$\mathbf{A}_{\text{full}}^{(t)} - \hat{\mathbf{A}}_{\text{s,q}}^{(t)} = (\mathbf{A}_{\text{s,q}}^{(t)} + \Delta_{\text{quant}}^{(t)}) - \hat{\mathbf{A}}_{\text{s,q}}^{(t)}$$
$$= (\mathbf{A}_{\text{s,q}}^{(t)} + \Delta_{\text{quant}}^{(t)}) - (\mathbf{A}_{\text{s,q}}^{(t)} + \Delta_{\text{quant}}^{(t_{\text{ref}})}) \quad (19)$$
$$= \Delta_{\text{quant}}^{(t)} - \Delta_{\text{quant}}^{(t_{\text{ref}})}.$$

Based on Proposition 3.3, we have:

$$\mathbb{E}_t \underbrace{\left[ \left\| \mathbf{A}_{\text{full}}^{(t)} - \tilde{\mathbf{A}}_{\text{s,q}}^{(t)} \right\|_F \right]}_{\text{second-order}} = \mathbb{E}_t \left[ \left\| \hat{\Delta}_{\text{quant}}^{(t)} - \hat{\Delta}_{\text{quant}}^{(t_{\text{ref}})} \right\|_F \right] \leq \mathbb{E}_t \left[ \left\| \Delta_{\text{quant}}^{(t)} - \Delta_{\text{quant}}^{(t_{\text{ref}})} \right\|_F \right] = \mathbb{E}_t \underbrace{\left[ \left\| \mathbf{A}_{\text{full}}^{(t)} - \hat{\mathbf{A}}_{\text{s,q}}^{(t)} \right\|_F \right]}_{\text{first-order}}.$$
$$(20)$$

Therefore, Theorem 3.4 holds.

## B    DETAILS OF SELECTED EVALUATION METRICS

### B.1    MULTI-ASPECTS METRICS EVALUATION

This evaluation suite includes absolute quality of videos and relative difference metrics that quantify the difference between FP16 generation.

**Absolute Quality.** Consistent with prior quantization works (Zhao et al., 2024; Feng et al., 2025c), we adopt CLIPSIM, VQA, and FlowScore to measure text-video alignment, quality, and temporal consistency, respectively.

**Relative Difference Metrics.** Following prior sparse attention works (Xi et al., 2025; Yuan et al., 2024; Ren et al., 2025; Zhang et al., 2025d), we adopt Peak Signal-to-Noise Ratio (PSNR), Structural Similarity Index Measure (SSIM), and Learned Perceptual Image Patch Similarity (LPIPS) for pixel-space differences, structural similarity, and high-level patch similarity, respectively.

All the evaluations are conducted on high-resolution generation tasks. Due to the computational overhead, we use the OpenSORA prompt sets used in (Zhao et al., 2024; Feng et al., 2025c) for video generation.

---

**Algorithm 1** QuantSparse: Calibration to Inference Pipeline

---

**Require:** Pre-trained video diffusion transformer $M$ (FP16), calibration dataset $\mathcal{D}_{cal}$, target bit-width (W/A), denoising steps $T$, cache interval $\tau$
**Ensure:** Quantized-sparse model $M_{QS}$, generated video $Y$

1: **Calibration Phase:**
2:   Initialize quantization parameters $\{s, z\}$ for weights (W) and activations (A)
3:   Input $X \in \mathcal{D}_{cal}$ to $M$
4:   Compute token saliency $s_j$ using Eq. 7 for FP model $M$
5:   Select top-$k$ salient tokens $I = \{j \mid s_j \text{ is top-}k\}$
6:   **Global Guidance Distillation:**
7:     Calculate $\mathcal{L}_{global}$ using Eq. 6
8:   **Local Guidance Distillation:**
9:     Calculate $\mathcal{L}_{local}$ using Eq. 8
10:   Optimize quantization parameters using Eq. 9 with $\mathcal{L}_{global}$ and $\mathcal{L}_{local}$
11:   Obtain quantized model $M_{quant}$ with optimized $\{s, z\}$
12: **Inference Phase:**
13:   Load $M_{quant}$ and input prompt $P$.
14:   Input $P$ into $M_{quant}$ and initialize cached residuals $\{\Delta_{quant}^{(t_{ref})}, \hat{\Delta}_{quant}^{(t_{ref})}\}$
15:   **for** t **in** $T$
16:     Compute quantized sparse attention:

$$A_{s,q}^{(t)} = \text{SparseAttention}(Q_{quant}, K_{quant}, V_{quant}; M)$$

17:     **if** $t - t_{ref} \leq \tau$
18:       Reuse cached residuals: $\Delta_{curr} = \Delta_{quant}^{(t_{ref})} + \hat{\Delta}_{quant}^{(t_{ref})}$
19:     **else**
20:       Update $t_{ref} = t$, recompute and cache residuals
21:     **endif**
22:     Refine attention using Eq. 16
23:   **endfor**
24:   Generate video $Y$ **return** $Y$

---

## B.2 BENCHMARK EVALUATION

To further provide benchmark evaluation, we follow previous works (Feng et al., 2025c; Zhao et al., 2024). We select 8 major dimensions from Vbench (Huang et al., 2024b), including frame-wise quality, temporal quality, and semantic evaluation.

For **Frame-wise Quality**, we select *Imaging Quality* and *Aesthetic Quality* for distortion assessment and artistic and beauty evaluation. For **Temporal Quality**, we use *Dynamic Degree*, *Motion Smoothness*, *Subject Consistency*, and *Background Consistency* for degree of dynamics, physical law smoothness, subject's appearance consistent, and temporal consistency of the background, respectively. For **Semantic Evaluation**, we use *Scene* and *Overall Consistency* for text prompt scene consistency and overall video-text consistency.

The evaluation follows the suite provided by VBench (Huang et al., 2024b). We generate one video for each prompt, same as previous works (Zhao et al., 2024; Feng et al., 2025c). Due to the large prompt sets used in VBench, we slightly decrease the resolution for computational efficiency. In addition, this experimental setup also provides an additional evaluation of multi-resolution video generation performance, which proves the generalization and effectiveness of our method in different application scenarios.

## C EXPERIMENT SETTINGS

Same with prior works (Zhao et al., 2024; Ashkboos et al., 2024; Feng et al., 2025c), we adopt channel-wise weight quantization and dynamic token-wise activation quantization. And we use uniform symmetry quantization for both weight and activation for better hardware acceleration and memory saving. For fair comparison, we apply the same quantization granularity for all quantization

Table 6: Text-to-Video generation experiments on more huge models.

| Method | #Bits (W/A) | Density↓ | Quality | | | | | |
|---|---|---|---|---|---|---|---|---|
| | | | *Video Quality Metrics* | | | *FP Diff. Metrics* | | |
| | | | CLIPSIM↑ | VQA↑ | ΔFSCore↓ | PSNR↑ | SSIM↑ | LPIPS↓ |
| HunyuanVideo 13B (CFG = 6.0, 720 × 1280p, frames = 60) | | | | | | | | |
| Full Prec. | 16/16 | 100% | 0.184 | 81.23 | 0.000 | - | - | - |
| SmoothQuant | 6/6 | 100% | 0.180 | 69.55 | 1.406 | 15.91 | 0.553 | 0.411 |
| QuaRot | 6/6 | 100% | 0.182 | 72.28 | 0.546 | 16.99 | 0.590 | 0.378 |
| ViDiT-Q | 6/6 | 100% | 0.182 | 72.36 | 0.937 | 18.24 | 0.623 | 0.335 |
| Q-VDiT | 6/6 | 100% | 0.182 | 73.68 | 1.232 | 21.02 | 0.675 | 0.264 |
| QuaRot+SVG | 6/6 | 25% | 0.181 | 72.57 | 0.718 | 16.85 | 0.581 | 0.385 |
| Q-VDiT+SVG | 6/6 | 25% | 0.181 | 72.59 | 1.405 | 20.38 | 0.658 | 0.284 |
| QuaRot+SVG | 6/6 | 15% | 0.181 | 72.60 | 0.997 | 16.85 | 0.578 | 0.394 |
| Q-VDiT+SVG | 6/6 | 15% | 0.181 | 72.04 | 1.763 | 19.94 | 0.644 | 0.307 |
| **QuantSparse** | 6/6 | 25% | **0.183** | 81.17 | 0.435 | **22.71** | **0.720** | **0.221** |
| **QuantSparse** | 6/6 | 15% | **0.183** | **82.26** | **0.328** | 22.68 | **0.720** | 0.224 |
| Wan2.1 14B (CFG = 5.0, 720 × 1280p, frames = 80) | | | | | | | | |
| Full Prec. | 16/16 | 100% | 0.182 | 90.79 | 0.000 | - | - | - |
| SmoothQuant | 6/6 | 100% | 0.178 | 62.25 | 0.363 | 13.06 | 0.404 | 0.656 |
| QuaRot | 6/6 | 100% | 0.180 | 66.56 | 0.313 | 13.59 | 0.409 | 0.566 |
| ViDiT-Q | 6/6 | 100% | 0.180 | 71.26 | 0.251 | 15.30 | 0.513 | 0.376 |
| Q-VDiT | 6/6 | 100% | 0.180 | 89.10 | 0.082 | 18.13 | 0.610 | 0.264 |
| QuaRot+SVG | 6/6 | 25% | 0.179 | 67.64 | 0.336 | 13.60 | 0.407 | 0.555 |
| Q-VDiT+SVG | 6/6 | 25% | 0.179 | 88.29 | 0.091 | 16.69 | 0.563 | 0.323 |
| QuaRot+SVG | 6/6 | 15% | 0.180 | 60.14 | 0.396 | 13.55 | 0.399 | 0.567 |
| Q-VDiT+SVG | 6/6 | 15% | 0.179 | 85.26 | 0.182 | 15.94 | 0.532 | 0.367 |
| **QuantSparse** | 6/6 | 25% | **0.182** | 89.96 | **0.002** | **18.67** | **0.622** | **0.240** |
| **QuantSparse** | 6/6 | 15% | 0.181 | **92.87** | 0.060 | **18.67** | 0.616 | 0.277 |

methods. We adopt channel-wise scale used in (Xiao et al., 2023a; Wu et al., 2024; Zhao et al., 2024; Feng et al., 2025c) and rotation-based matrix used in (Ashkboos et al., 2024; Zhao et al., 2024; Sun et al., 2024b) for quantization. We follow block-wise post-training strategy used in (Wu et al., 2024; Chen et al., 2024; Sun et al., 2024b) for calibration. All the experiments are conducted on a single NVIDIA A800 GPU.

During calibration, we set channel-wise scale, rotation matrix, and quantization scale as learnable following (Feng et al., 2025c; Sun et al., 2024b). We use 20 random generated samples and train 15 epoch for each transformer block. We apply the same calibration samples and epochs for all methods for fair comparison. We use AdamW (Loshchilov & Hutter, 2017) optimizer and cosine learning rate scheduler. For the channel-wise scale and rotation matrix, we use a learning rate of $5e^{-3}$. For the learnable quantization scale, we use a learning rate of $5e^{-2}$. For distillation, we use $r = 128$ for global distillation pooling, $k = 256$ for salient query selection, and $\lambda_{\text{global}} = 1e^{-4}, \lambda_{\text{global}} = 1e^{-4}$ for Wan2.1-1.3B, Wan2.1-14B, and $\lambda_{\text{global}} = 1.0, \lambda_{\text{global}} = 1e^2$ for HunyuanVideo, respectively. The selection of distillation balancing factor is based on the order of magnitude of the loss. For sparse attention, we use a fixed cache refreshing interval of 5, and use $k = 16$ for SVD.

For deployment, we quantize the weight and absorb all the quantization parameters following (Zhao et al., 2024; Sun et al., 2024b; Feng et al., 2025c; Ashkboos et al., 2024). For activation, we use dynamic online quantization same as (Feng et al., 2025c; Sun et al., 2024b; Zhao et al., 2024).

## D MORE EVALUATION RESULTS ON WAN2.1-1.3B

We present comprehensive evaluation on Wan2.1-1.3B (Wan et al., 2025) in Tab. 6. Since Wan2.1-1.3B has less computation budget and we find that it will suffer from serious performance degradation under high sparsity, we uniformly adopt 40 density in sparse attention to ensure its performance.

Different quantization methods have obvious performance degradation, especially under W4A8. Among them, the quantization method specially designed for video model Q-VDiT (Feng et al.,

Table 7: Performance of text-to-video generation under VBench evaluation benchmark suite. We evaluate on Imaging Quality (IQ), Aesthetic Quality (AQ), Motion Smoothness (MS), Dynamic Degree (DD), Background Consistency (BC), Subject Consistency (SuC), Scene Consistency (ScC), and Overall Consistency (OC). Higher (↑) metrics represent better performance. **Bold**: the best result. Underline: The second best result.

| Method | #Bits (W/A) | Density | IQ | AQ | MS | DD | BC | SuC | ScC | OC |
|---|---|---|---|---|---|---|---|---|---|---|
| Wan2.1 1.3B (CFG = 6.0, 480 × 832p, frames = 80) | | | | | | | | | | |
| Full Prec. | 16/16 | 100% | 64.05 | 57.86 | 97.03 | 87.50 | 94.94 | 93.00 | 16.72 | 23.16 |
| QuaRot+SVG | 6/6 | 40% | 62.53 | 52.16 | 95.48 | 81.94 | 93.65 | 89.20 | 12.43 | 22.42 |
| Q-VDiT+SVG | 6/6 | 40% | 64.01 | 53.89 | 96.25 | 81.94 | 94.23 | 91.78 | 17.81 | 22.90 |
| **QuantSparse** | 6/6 | 40% | **64.96** | **56.44** | **96.68** | **83.33** | **94.84** | **92.56** | **18.46** | **23.12** |
| QuaRot+SVG | 4/8 | 40% | 54.45 | 43.60 | 96.29 | 73.61 | 94.99 | 87.02 | 8.14 | 18.88 |
| Q-VDiT+SVG | 4/8 | 40% | 56.08 | 48.12 | 97.27 | 61.11 | 95.86 | 89.72 | 10.32 | 19.89 |
| **QuantSparse** | 4/8 | 40% | **64.41** | **58.00** | **97.35** | **87.50** | **94.99** | **93.02** | **18.24** | **23.31** |
| HunyuanVideo 13B (CFG = 6.0, 512 × 768p, frames = 60) | | | | | | | | | | |
| Full Prec. | 16/16 | 100% | 62.30 | 62.49 | 99.00 | 56.94 | 98.08 | 95.30 | 33.36 | 26.85 |
| QuaRot+SVG | 6/6 | 25% | 56.82 | 57.23 | 97.93 | 40.00 | 97.75 | 95.10 | 23.98 | 25.63 |
| Q-VDiT+SVG | 6/6 | 25% | 59.22 | 58.77 | 97.96 | 40.00 | 97.60 | 95.68 | 26.80 | 25.87 |
| QuaRot+SVG | 6/6 | 15% | 53.95 | 56.43 | 97.84 | 38.89 | 97.48 | 94.40 | 23.36 | 25.57 |
| Q-VDiT+SVG | 6/6 | 15% | 57.43 | 58.02 | 97.84 | 38.61 | 97.07 | 95.20 | 24.27 | 25.74 |
| **QuantSparse** | 6/6 | 25% | **60.24** | **59.06** | **99.01** | **43.06** | **98.33** | **96.06** | **28.42** | 26.62 |
| **QuantSparse** | 6/6 | 15% | 59.54 | 58.87 | 98.95 | 40.28 | 98.08 | 95.84 | 27.69 | **26.63** |
| QuaRot+SVG | 4/8 | 25% | 45.81 | 56.59 | 98.26 | 22.22 | 98.18 | 95.78 | 21.00 | 24.64 |
| Q-VDiT+SVG | 4/8 | 25% | 44.94 | 56.62 | 98.36 | 23.61 | 97.98 | 96.06 | 18.53 | 24.81 |
| QuaRot+SVG | 4/8 | 15% | 43.51 | 55.35 | 98.21 | 20.83 | 97.21 | 95.15 | 18.31 | 24.50 |
| Q-VDiT+SVG | 4/8 | 15% | 42.16 | 55.32 | 98.32 | 20.83 | 97.96 | 95.48 | 16.64 | 24.68 |
| **QuantSparse** | 4/8 | 25% | **59.85** | **59.37** | **99.08** | 38.89 | **98.32** | **96.41** | 29.80 | **26.92** |
| **QuantSparse** | 4/8 | 15% | 59.27 | 59.20 | 99.04 | **40.28** | 98.21 | 96.18 | **30.31** | **26.92** |
| Wan2.1 14B (CFG = 5.0, 480 × 832p, frames = 80) | | | | | | | | | | |
| Full Prec. | 16/16 | 100% | 63.38 | 59.56 | 96.73 | 86.11 | 96.71 | 90.84 | 28.13 | 25.68 |
| QuaRot+SVG | 6/6 | 25% | 61.77 | 54.13 | 95.89 | 45.83 | 94.78 | 90.20 | 17.59 | 23.37 |
| Q-VDiT+SVG | 6/6 | 25% | 60.92 | 57.53 | 96.44 | 82.50 | 95.48 | 89.34 | 27.76 | 25.46 |
| QuaRot+SVG | 6/6 | 15% | 61.42 | 54.09 | 95.78 | 45.83 | 94.70 | 89.95 | 16.50 | 23.08 |
| Q-VDiT+SVG | 6/6 | 15% | 59.77 | 56.56 | 96.31 | 82.50 | 95.68 | 89.25 | 27.05 | 25.28 |
| **QuantSparse** | 6/6 | 25% | **63.89** | **58.77** | **96.77** | 84.72 | **96.48** | **90.91** | **29.80** | 25.80 |
| **QuantSparse** | 6/6 | 15% | 63.87 | 58.32 | 96.69 | **90.28** | 96.29 | 90.85 | 29.14 | **26.07** |
| QuaRot+SVG | 4/8 | 25% | 62.53 | 57.24 | 96.52 | 87.50 | 95.40 | 89.77 | 22.69 | 25.11 |
| Q-VDiT+SVG | 4/8 | 25% | 60.92 | 58.53 | 96.44 | 87.50 | 95.48 | 89.34 | 22.76 | 25.46 |
| QuaRot+SVG | 4/8 | 15% | 61.16 | 56.71 | 96.32 | 87.50 | 95.33 | 89.67 | 22.14 | 25.05 |
| Q-VDiT+SVG | 4/8 | 15% | 59.77 | 57.56 | 96.31 | 87.50 | 95.68 | 89.25 | 22.04 | 25.08 |
| **QuantSparse** | 4/8 | 25% | 63.55 | **59.59** | **96.82** | **87.50** | **96.69** | **90.69** | **27.76** | 25.81 |
| **QuantSparse** | 4/8 | 15% | **63.81** | 58.86 | 96.75 | **87.50** | 96.56 | 90.55 | 26.09 | **25.93** |

2025c) and the strong LLM quantization method Quarot (Ashkboos et al., 2024) show relatively stronger performance. For a broader and fair comparison, we add existing sparse attention methods to Q-VDiT and Quarot to verify the effect of naive combination of model quantization and sparse attention. We find that when combining Q-VDiT and Quarot with different sparse attention methods, the performance decreases to varying degrees, and the performance of SVG (Xi et al., 2025) decreases the least. Therefore, we chose SVG as our baseline sparse attention in all other experiments.

Compared with all existing methods, QuantSparse achieves SOTA performance under all bit settings, and is almost lossless compared with the FP model. It is worth mentioning that QuantSparse even surpasses all quantization-only methods. It not only achieves better compression effect, but also has better performance, which fully demonstrates the effectiveness of our method.

## E  VBENCH EVALUATION RESULTS

We present the VBench (Huang et al., 2024b) evaluation results in Tab. 7. Under the comprehensive evaluation of all 8 dimensions, the naive combination of Q-VDiT (Feng et al., 2025c), Quarot (Ashkboos et al., 2024) and SVG (Xi et al., 2025) all show significant performance degradation, which fully demonstrates the disadvantage of simply combining existing quantization and sparse attention methods. While QuantSparse achieves comprehensive SOTA performance in all bit settings of all models, and is almost lossless compared with FP model, even better in some dimensions. For Wan2.1-14B (Wan et al., 2025) under W4A8, QuantSparse achieves 63.55 and 63.81 under 25% and 15% attention density, respectively, surpassing 63.38 of FP model.

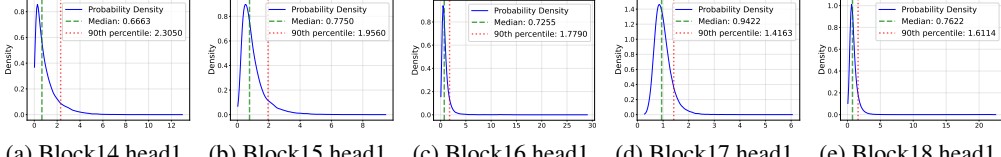

(a) Block14 head1.  (b) Block15 head1.  (c) Block16 head1.  (d) Block17 head1.  (e) Block18 head1.

Figure 6: More token saliency distribution of Wan2.1-1.3B (Wan et al., 2025).

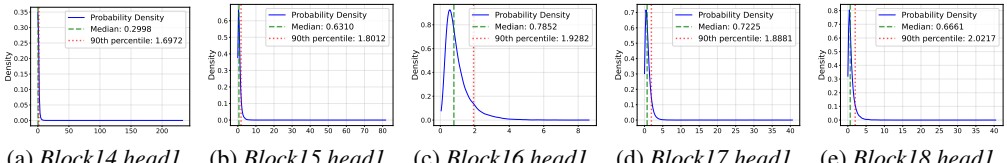

(a) *Block14 head1*.  (b) *Block15 head1*.  (c) *Block16 head1*.  (d) *Block17 head1*.  (e) *Block18 head1*.

Figure 7: More token saliency distribution of HunyuanVideo-13B (Kong et al., 2024).

## F  MORE ANALYSIS OF MULTI-SCALE SALIENT ATTENTION DISTILLATION

We present more analysis of the proposed Multi-Scale Salient Attention Distillation (MSAD) here.

We conducted quantitative experiments to test the impact of quantization and sparsification on attention shift by measuring the attention Mean Square Error (MSE). The results are collected from 1000 random samples on Wan2.1-1.3B (Wan et al., 2025) under W4A8 and 40% attention density. The results are presented in Tab. 8. The attention shift caused by the simple combination of quantization and sparsification methods is much greater than the sum of individual shifts. This proves the joint effect of quantization and sparsification on attention error, and our core motivation "amplified attention shift".

Table 8: Quantitative experiment on attention shift caused by different compression techniques.

| Method | Attention Shift |
|---|---|
| Quantization (QuaRot (Ashkboos et al., 2024)) | 0.216 |
| Sparsification (SVG (Xi et al., 2025)) | 0.134 |
| **Quantization+Sparsification** | **0.685** |

We supplement 4 additional attention map comparisons in Fig. 8 and Fig. 9, showing the attention distribution difference between the FP model and quantized model. The results are collected from Wan2.1-1.3B under W4A8.

Each column in Fig. 8 and Fig. 9 corresponds to the attention difference between the same attention map before and after the proposed distillation MSAD. This indicates that our MSAD effectively alleviates the attention shift.

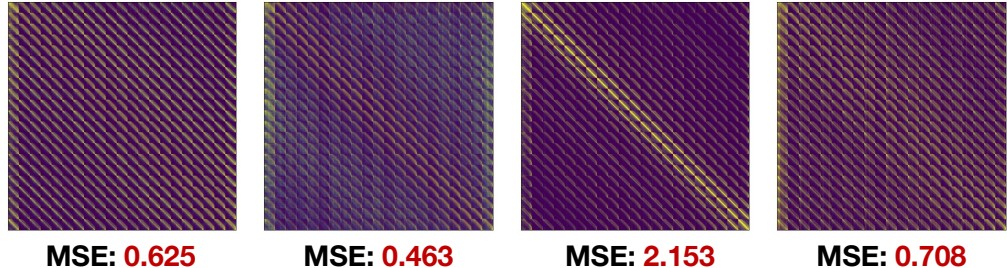

Figure 8: Attention differences between FP model and quantized model without distillation.

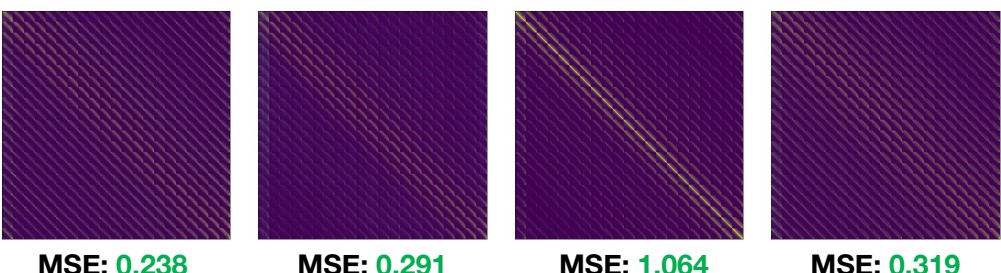

Figure 9: Attention differences between FP model and quantized model with distillation.

We present more visualization of heavy-tail token saliency distribution in Fig. 6 and Fig. 7. It can be seen that a significantly heavy-tailed token saliency phenomenon appears in different blocks of Wan2.1 (Wan et al., 2025) and HunyuanVideo (Kong et al., 2024), which fully shows that our salient local distillation is meaningful.

To further prove the effect of top-$k$ salient queries selection, we compare with random selection methods and present the results in Tab. 9. Compared with random selection, our top-$k$ salient selection further improves the PSNR from 15.49 to 16.82, fully demonstrating the effectiveness of our local distillation.

Table 9: Ablation results of local distillation.

| Method | VQA↑ | PSNR↑ | SSIM↑ | LPIPS↓ |
|--------|------|-------|-------|--------|
| None | 81.92 | 14.35 | 0.486 | 0.425 |
| *Random* | 83.17 | 15.49 | 0.523 | 0.372 |
| **Salient** | **86.95**$_{+5.03}$ | **16.82**$_{+2.47}$ | **0.561**$_{+0.075}$ | **0.325**$_{-0.100}$ |

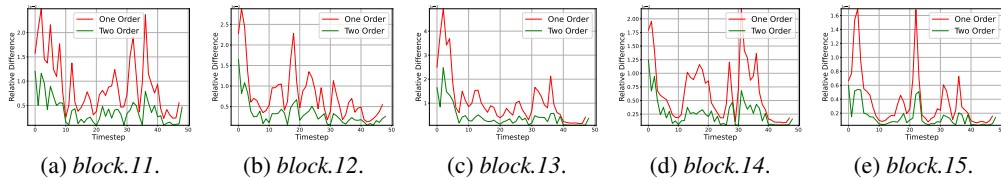

(a) *block.11*.  (b) *block.12*.  (c) *block.13*.  (d) *block.14*.  (e) *block.15*.

Figure 10: More residual temporal difference distribution of HunyuanVideo-13B (Kong et al., 2024).

## G    MORE ANALYSIS OF SECOND-ORDER SPARSE ATTENTION REPARAMETERIZATION

We present more analysis of the proposed Second-Order Sparse Attention Reparameterization (SSAR) here. We present more visualization of residual temporal difference in Fig. 10. It can be seen that after the introduction of quantization, the numerical difference of the first-order residuals of adjacent time steps cannot be simply ignored. However, the numerical difference of the second-order residual is significantly smaller than that of the first-order residual, so the use of the second-order residual has a better approximation effect.

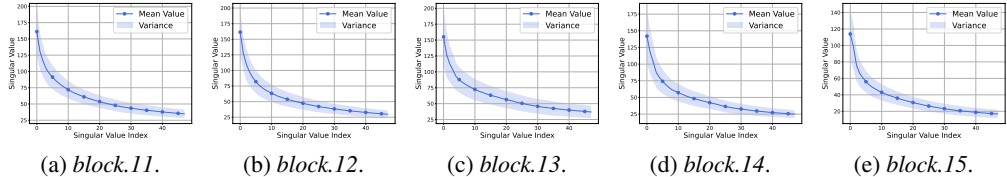

(a) *block.11.*  (b) *block.12.*  (c) *block.13.*  (d) *block.14.*  (e) *block.15.*

Figure 11: More singular value distribution of all timesteps of HunyuanVideo-13B (Kong et al., 2024).

To verify the motivation of using the temporal-stable component of the second-order residual, we visualize more singular value distribution of all timesteps in Fig. 11. It can be seen that in different blocks of different models, the second-order residuals at different time steps show considerable stability. Therefore, the second-order residual after SVD can retain the characteristics of time stability, further reduce the variance caused by different time steps, and have better approximation effect.

We further visualize more attention error comparison in Fig. 12. It can be seen that the residual mechanism significantly reduces the attention error, which proves the importance of sparse attention reparameterization. At the same time, compared with the first-order residual, the second-order residual further reduces the attention error, which proves the necessity of introducing the second-order residual after quantization. Also, the second-residual after using SVD can further reduce the attention error, which proves that we have indeed extracted the temporally stable component and achieved the best attention approximation effect.

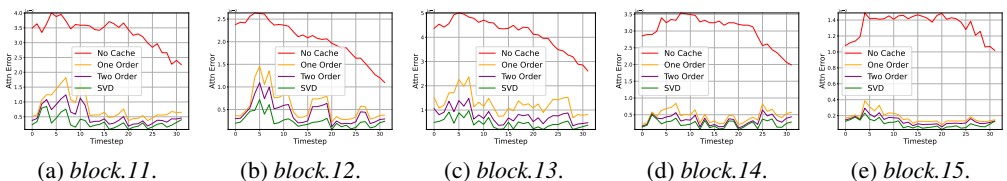

(a) *block.11.*  (b) *block.12.*  (c) *block.13.*  (d) *block.14.*  (e) *block.15.*

Figure 12: More attention error comparison of HunyuanVideo-13B (Kong et al., 2024).

## H   MORE ABLATION STUDY

Here, we provide more ablation study about the proposed Multi-Scale Salient Attention Distillation (MSAD) and Second-Order Sparse Attention Reparameterization (SSAR).

We first study the pooling stride $s$ used in Eq. 6 and salient token $k$ in Eq. 8 to verify the hyperparameter selection of both global and local distillation. We present the results in Tab. 10. It can be seen that different hyperparameters can improve the distillation performance. This shows that our distillation method is both effective and robust, which is insensitive to hyperparameters. This also demonstrates that the memory-efficient distillation are effective enough and we do not have to use the giant complete attention map to supervise the attention module. Higher $s$ and lower $k$ can reduce memory, but typically harm performance. Yet we identify that decreasing $s$ and increasing $k$ also brings little improvement. Since $s = 128$ and $k = 256$ are both effective and efficient as shown in Fig. 3d, we choose this balanced selection.

Table 10: Ablation on $s$ and $k$ used in attention distillation.

| Value | VQA↑ | PSNR↑ | SSIM↑ | LPIPS↓ |
|---|---|---|---|---|
| None | 81.92 | 14.35 | 0.486 | 0.425 |
| pooling stride $s$ | | | | |
| s=64 | 85.19 | 16.05 | 0.543 | 0.348 |
| **s=128** | 85.26 | 16.01 | 0.547 | 0.349 |
| s=256 | 85.12 | 15.93 | 0.545 | 0.355 |
| salient token $k$ | | | | |
| k=128 | 86.21 | 16.72 | 0.551 | 0.349 |
| **k=256** | 86.95 | 16.82 | 0.561 | 0.325 |
| k=512 | 86.93 | 16.95 | 0.561 | 0.324 |

We then study the top-$r$ components in SVD used in Eq. 16, and present the results in Tab. 11. Compared with the original second-order residual, it can be seen that the different selection of $r$ in SVD can improve the temporal stability of the second-order residual and bring better performance. In our experiment, we chose $r = 16$ as it achieves good performance. We further explore higher-order residual effectiveness on attention approximation. Compared with the Second-Order residual, Third-

Table 11: Ablation on SVD used in SSAR.

| Method | VQA$_\uparrow$ | PSNR$_\uparrow$ | SSIM$_\uparrow$ | LPIPS$_\downarrow$ |
|--------|------|------|------|-------|
| None | 68.00 | 14.16 | 0.470 | 0.445 |
| *First* | 70.82 | 17.08 | 0.572 | 0.285 |
| *Second* | 89.73 | 18.68 | 0.616 | 0.258 |
| *Third* | 89.71 | 18.70 | 0.620 | 0.263 |
| *top-8* | 91.12 | 18.69 | 0.621 | 0.253 |
| *top-16* | **91.98**$_{+23.98}$ | **18.72**$_{+4.56}$ | **0.630**$_{+0.160}$ | **0.240**$_{-0.205}$ |
| *top-32* | 91.75 | 18.72 | 0.628 | 0.242 |

Order residual only slightly improves PSNR from 18.70 to 18.68 and decreases the performance on VQA, SSIM, and LPIPS. This indicates that the stability brought by higher-order residuals will gradually saturate, and we attribute it to the additional noise brought by longer time series information on higher-order residuals. The second-order residual not only stabilizes the first-order residual, but SVD can further reduce spatiotemporal noise.

We then study the weight factor used in Eq. 9 to verify the distillation robustness of hyperparameters. We present the results in Tab. 12. The values are selected by controlling the distillation term to be of the same order of magnitude as $\mathcal{L}_{\text{quant}}$. It can be seen that different weight factors improve the model performance. This shows that our distillation method is not only effective but also insensitive to the choice of hyperparameters, indicating its generalization and effectiveness. Since $\lambda_{\text{global}} = 1e-4$ and $\lambda_{\text{local}} = 1e-4$ are good enough, and the hyperparameter selection is robust, we do not further fine-tune the hyperparameter selection.

Table 12: Ablation on $\lambda_*$ used in Eq. 9.

| $\lambda_*$ | VQA$_\uparrow$ | PSNR$_\uparrow$ | SSIM$_\uparrow$ | LPIPS$_\downarrow$ |
|------|------|------|------|------|
| None | 81.92 | 14.35 | 0.486 | 0.425 |
| | | * = global | | |
| 5e-3 | 84.76 | 15.79 | 0.518 | 0.362 |
| **1e-4** | 85.26 | **16.01** | **0.547** | **0.349** |
| 5e-4 | **85.33** | 15.72 | 0.540 | 0.351 |
| | | * = local | | |
| 5e-3 | 86.73 | **16.86** | 0.547 | 0.336 |
| **1e-4** | **86.95** | 16.82 | 0.561 | **0.325** |
| 5e-4 | 86.54 | 16.72 | **0.562** | 0.328 |

We further compare our Multi-Scale Salient Attention Distillation (MSAD) with full-attention distillation (using the complete FP attention map as the target) on Wan2.1-1.3B (Wan et al., 2025) under W4A8 quantization. The results are shown in Tab. 13. MSAD achieves nearly identical performance to full-attention distillation. The results highlight MSAD's efficiency advantages while maintaining comparable performance.

Table 13: Ablation study on full-attention distillation.

| Method | Resolution | PSNR$_\uparrow$ | LPIPS$_\downarrow$ | Attention Memory Cost (GB)$_\downarrow$ | Calibration Time (Hours)$_\downarrow$ |
|--------|-----------|------|-------|------|------|
| | | | `Wan2.1 1.3B` | | |
| Full Attention | (17472, 17472) | 15.25 | 0.338 | 6.82 | 1.86 |
| **MSAD** ($s = 64$) | (273, 273) | 15.23 | 0.338 | 0.17 | 0.66 |
| **MSAD** ($s = 128$) | (137, 137) | 15.22 | 0.338 | 0.14 | 0.64 |
| **MSAD** ($s = 256$) | (69, 69) | 15.21 | 0.339 | 0.13 | 0.63 |

**Effect of attention density.** We conduct an ablation study on attention density, analyzing the trade-off between performance and inference speed. The results are presented in Tab. 5. As shown, a 25% density offers a good balance, achieving a significant 1.55× speedup with minimal performance degradation (PSNR of 18.72). A 15% density further boosts the speedup to 1.74× while maintaining acceptable performance (PSNR of 18.22). Based on these results, we selected 25% and 15% density for the experiments presented in the main paper. The 25% density provides a strong baseline for high performance with good acceleration, while the 15% density demonstrates the potential for even greater inference speedup at a slightly decreased performance trade-off.

# I    CALIBRATION COMPUTATION RESOURCE

We study the calibration computation resource of each of our proposed methods and the overall pipeline. As Second-Order Sparse Attention Reparameterization (SSAR) is used for only inference, for calibration, we only add Multi-Scale Salient Attention Distillation (MSAD) compared to naive Post-Training Quantization (PTQ). We present the calibration resource in Tab. 14. Compared with

Table 14: Calibration computation resource report. PTQ denotes naive Post-Training Quantization without attention distillation.

| Method | Calibration Overload | | Performance | |
|---|---|---|---|---|
| | GPU Memory (GB)$\downarrow$ | GPU Time (Hours)$\downarrow$ | PSNR$\uparrow$ | LPIPS$\downarrow$ |
| Wan2.1 1.3B | | | | |
| PTQ | 16.21 | 0.62 | 10.57 | 0.587 |
| +*Global* | $16.27_{+0.4\%}$ | $0.63_{+0.2\%}$ | 13.27 | 0.452 |
| +*Local* | $16.28_{+0.4\%}$ | $0.63_{+0.2\%}$ | 13.85 | 0.421 |
| **QuantSparse** | $16.34_{+0.8\%}$ | $0.64_{+1.6\%}$ | $\mathbf{15.22}_{+4.65}$ | $\mathbf{0.338}_{-0.249}$ |
| HunyuanVideo 13B | | | | |
| PTQ | 39.22 | 5.08 | 16.27 | 0.472 |
| +*Global* | $39.33_{+0.3\%}$ | $5.10_{+0.4\%}$ | 18.42 | 0.357 |
| +*Local* | $39.32_{+0.3\%}$ | $5.11_{+0.6\%}$ | 18.96 | 0.342 |
| **QuantSparse** | $39.41_{+0.5\%}$ | $5.13_{+1.0\%}$ | $\mathbf{20.86}_{+4.59}$ | $\mathbf{0.272}_{-0.200}$ |
| Wan2.1 14B | | | | |
| PTQ | 47.39 | 2.57 | 14.35 | 0.425 |
| +*Global* | $47.54_{+0.3\%}$ | $2.59_{+0.8\%}$ | 16.01 | 0.349 |
| +*Local* | $47.50_{+0.2\%}$ | $2.58_{+0.4\%}$ | 16.82 | 0.325 |
| **QuantSparse** | $47.65_{+0.5\%}$ | $2.60_{+1.1\%}$ | $\mathbf{18.72}_{+4.37}$ | $\mathbf{0.240}_{-0.185}$ |

naive PTQ, our *Global Distillation* only brings an average of 0.8% extra time burden and almost no additional memory consumption because of its efficient low-resolution attention operation. Also, our *Local Distillation* only needs to calculate the token saliency distribution once before each block calibration and reuse the salient token index in each optimization iteration, which is also very efficient. These two distillation methods are not only efficient but also can effectively alleviate the attention shift caused by quantization and improve the video generation effect. QuantSparse has significantly improved the model performance by combining two distillation methods, while ensuring high efficiency.

To further prove the effectiveness of proposed Second-Order Sparse Attention Reparameterization (SSAR), we present the inference burden brought by SSAR in Tab. 15. Compared with Non-Reparameterization, the cache-based method only requires one additional matrix addition operation for the sparse attention output, which is very efficient. Therefore, the cache-based method will only bring little additional latency and memory burden. Furthermore, the second-order residual can store and calculate the second-order term and the first-order term together. Therefore, compared with the first-order residual, the second-order residual only requires an additional second order term calculation, but significantly improves the sparse attention performance under quantization, and improves the PSNR from 17.08 to 18.68 under Wan2.1-14B (Wan et al., 2025). In addition, using SVD to extract the temporally stable component of second-order residuals brings almost no additional consumption, but can further improve the effect of second-order residuals, which further decreases LPIPS from 0.258 to 0.240 under Wan2.1-14B.

## J  COMBINATION WITH OTHER ACCELERATION TECHNIQUES

To further validate the integration ability of QuantSparse with other acceleration techniques, we combined it with existing attention quantization techniques SageAttention (Zhang et al., 2024b) and cache techniques TeaCache (Liu et al., 2024a), and presented the results in Tab. 16. All the experiments are conducted on Wan2.1-14B (Wan et al., 2025) under W4A8 quantization setting. We apply SageAttention by quantizing attention into 8-bit. For TeaCache, we set the threshold as 0.16 to ensure performance.

It can be seen that, despite retaining only 15% attention density under W4A8 quantization, the combination of QuantSparse and SageAttention still incurs almost no performance loss. This indicates that QuantSparse is highly friendly to sparsification and quantization, fully demonstrating the necessity of attention distillation and second-order reparameterization. Although further adding TeaCache may result in a slight performance decrease, it can bring significant additional inference acceleration. This provides a further trade-off between performance and inference speed, and also demonstrates the effectiveness of combining QuantSparse with cache-based methods.

Table 15: Sparse attention reparameterization resource report. 'None' denotes Non-Reparameterization.

| Method | Inference Overload | | Performance | |
|--------|--------------------|---|-------------|---|
| | GPU Memory (GB)$\downarrow$ | DiT Time (s)$\downarrow$ | PSNR$\uparrow$ | LPIPS$\downarrow$ |
| Wan2.1 1.3B | | | | |
| None | 5.44 | 312 | 10.57 | 0.587 |
| +*First* | 5.84$_{+7\%}$ | 313$_{+0.3\%}$ | 12.76 | 0.493 |
| +*Second* | 5.93$_{+9\%}$ | 313$_{+0.3\%}$ | 13.55 | 0.427 |
| **QuantSparse** | 5.93$_{+9\%}$ | 313$_{+0.3\%}$ | **15.22**$_{+4.65}$ | **0.338**$_{-0.249}$ |
| HunyuanVideo 13B | | | | |
| None | 24.34 | 725 | 16.27 | 0.472 |
| +*First* | 26.51$_{+9\%}$ | 729$_{+0.6\%}$ | 18.25 | 0.381 |
| +*Second* | 27.02$_{+11\%}$ | 730$_{+0.7\%}$ | 19.03 | 0.317 |
| **QuantSparse** | 27.02$_{+11\%}$ | 731$_{+0.8\%}$ | **20.86**$_{+4.59}$ | **0.272**$_{-0.200}$ |
| Wan2.1 14B | | | | |
| None | 26.04 | 2589 | 14.16 | 0.445 |
| +*First* | 27.86$_{+7\%}$ | 2593$_{+0.2\%}$ | 17.08 | 0.285 |
| +*Second* | 28.14$_{+8\%}$ | 2594$_{+0.2\%}$ | 18.68 | 0.258 |
| **QuantSparse** | 28.14$_{+8\%}$ | 2594$_{+0.2\%}$ | **18.72**$_{+4.56}$ | **0.240**$_{-0.205}$ |

We further provide more visualization results in Fig. 13. It can be seen that the combination of QuantSparse and other acceleration techniques not only shows almost no decrease in metrics but also maintains good visual effects without producing any decrease in visual quality.

Table 16: More efficiency comparison under W4A8 quantization setting. Sage. denotes SageAttention (Zhang et al., 2024b). Tea. denotes TeaCache (Liu et al., 2024a).

| Method | | | Density$\downarrow$ | Quality | | | Latency & Speed | |
|--------|--|--|---------|---------|--|--|-----------------|--|
| QuantSparse | SageAttention | TeaCache | | CLIPSIM$\uparrow$ | VQA$\uparrow$ | $\Delta$FSCore$\downarrow$ | DiT Time$\downarrow$ | Speedup$\uparrow$ |
| Wan2.1 14B (CFG = 5.0, 720 × 1280$p$, frames = 80) | | | | | | | | |
| | Full Prec. | | 100% | 0.182 | 90.79 | 0.000 | 4031s | 1.00× |
| ✓ | | | | 0.183 | 91.98 | 0.056 | 2594s | 1.55× |
| ✓ | ✓ | | 25% | 0.181 | 91.70 | 0.240 | 2480s | 1.63× |
| ✓ | ✓ | ✓ | | 0.180 | 84.01 | 0.211 | 1802s | 2.24× |
| ✓ | | | | 0.182 | 90.73 | 0.042 | 2315s | 1.74× |
| ✓ | ✓ | | 15% | 0.180 | 90.58 | 0.046 | 2201s | 1.83× |
| ✓ | ✓ | ✓ | | 0.179 | 86.24 | 0.249 | 1629s | 2.47× |

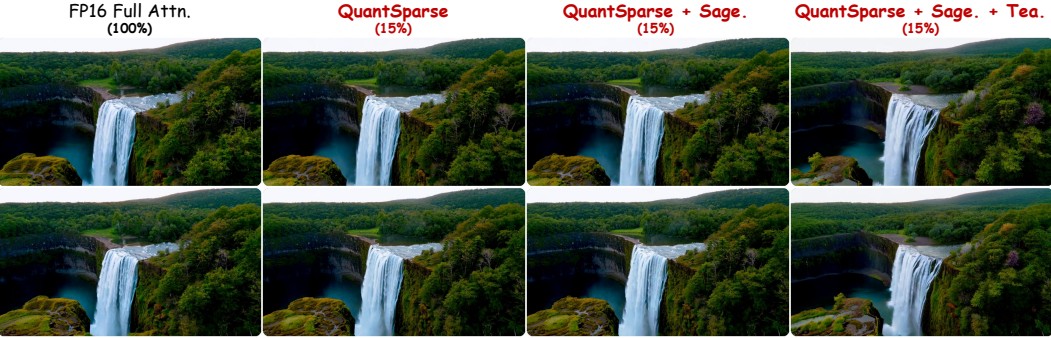

Figure 13: Combining with other acceleration techniques visualization on Wan2.1-14B under W4A8 quantization setting.

## K  IMAGE GENERATION EXPERIMENT

QuantSparse is designed as a general framework for Diffusion Transformers (DiTs) and is not limited to video generation. To validate its generalizability, we conducted an experiment on Hunyuan-DiT (Li et al., 2024c), a 1.5B parameters model targeting image generation. We evaluate on Draw-Bench (Saharia et al., 2022) under W4A8 quantization and present the results in Tab. 17. Even for image-generation DiTs, QuantSparse outperforms SOTA quantization baselines QuaRot (Ashkboos et al., 2024) and Q-VDiT (Feng et al., 2025c) while using only 40% attention density. This confirms that our framework generalizes to DiT-based visual generation tasks and not limited to video generation.

Table 17: Image generation experiment results on Hunyuan-DiT.

| Method | Density↓ | PSNR↑ | SSIM↑ | LPIPS↓ |
|---|---|---|---|---|
| | Hunyuan-DiT | | | |
| QuaRot | 100% | 17.30 | 0.627 | 0.460 |
| Q-VDiT | 100% | 19.32 | 0.658 | 0.347 |
| **QuantSparse** | 40% | **20.34** | **0.692** | **0.289** |

## L  THE USE OF LARGE LANGUAGE MODELS (LLMS)

In this paper, Large Language Models are only used as general-purpose auxiliary tools, primarily for document-level auxiliary tasks such as grammar checking and expression refinement. LLMs did not participate in the core conceptualization, method derivation, or experimental design of this research, nor did they contribute to any core writing content.

## M  MORE VISUAL COMPARISON

In the following pages, we provide more visual comparisons of different-scale video-generation models. 'Full Prec.' denotes the Full Precision model. (xx%) denotes the attention density.

We also provide the used text prompt for each figure:

1. Fig. 14: *A soaring drone footage captures the majestic beauty of a coastal cliff, its red and yellow stratified rock faces rich in color and against the vibrant turquoise of the sea. Seabirds can be seen taking flight around the cliff's precipices. As the drone slowly moves from different angles, the changing sunlight casts shifting shadows that highlight the rugged textures of the cliff and the surrounding calm sea. The water gently laps at the rock base and the greenery that clings to the top of the cliff, and the scene gives a sense of peaceful isolation at the fringes of the ocean. The video captures the essence of pristine natural beauty untouched by human structures.*

2. Fig. 15: *A serene night scene in a forested area. The first frame shows a tranquil lake reflecting the star-filled sky above. The second frame reveals a beautiful sunset, casting a warm glow over the landscape. The third frame showcases the night sky, filled with stars and a vibrant Milky Way galaxy. The video is a time-lapse, capturing the transition from day to night, with the lake and forest serving as a constant backdrop. The style of the video is naturalistic, emphasizing the beauty of the night sky and the peacefulness of the forest.*

3. Fig. 16: *A serene underwater scene featuring a sea turtle swimming through a coral reef. The turtle, with its greenish-brown shell, is the main focus of the video, swimming gracefully towards the right side of the frame. The coral reef, teeming with life, is visible in the background, providing a vibrant and colorful backdrop to the turtle's journey. Several small fish, darting around the turtle, add a sense of movement and dynamism to the scene. The video is shot from a slightly elevated angle, providing a comprehensive view of the turtle's surroundings. The overall style of the video is calm and peaceful, capturing the beauty and tranquility of the underwater world.*

4. Fig. 17: *The video captures the majestic beauty of a waterfall cascading down a cliff into a serene lake. The waterfall, with its powerful flow, is the central focus of the video. The surrounding landscape is lush and green, with trees and foliage adding to the natural beauty of the scene. The camera angle provides a bird's eye view of the waterfall, allowing viewers to appreciate the full height and grandeur of the waterfall. The video is a stunning representation of nature's power and beauty.*

5. Fig. 18: *The dynamic movement of tall, wispy grasses swaying in the wind. The sky above is filled with clouds, creating a dramatic backdrop. The sunlight pierces through the clouds, casting a warm glow on the scene. The grasses are a mix of green and brown, indicating a change in seasons. The overall style of the video is naturalistic, capturing the beauty of the landscape in a realistic manner. The focus is on the grasses and their movement, with the sky serving as a secondary element. The video does not contain any human or animal elements.*

6. Fig. 19: *The video captures the majestic beauty of a waterfall cascading down a cliff into a serene lake. The waterfall, with its powerful flow, is the central focus of the video. The surrounding landscape is lush and green, with trees and foliage adding to the natural beauty of the scene. The camera angle provides a bird's eye view of the waterfall, allowing viewers to appreciate the full height and grandeur of the waterfall. The video is a stunning representation of nature's power and beauty.*

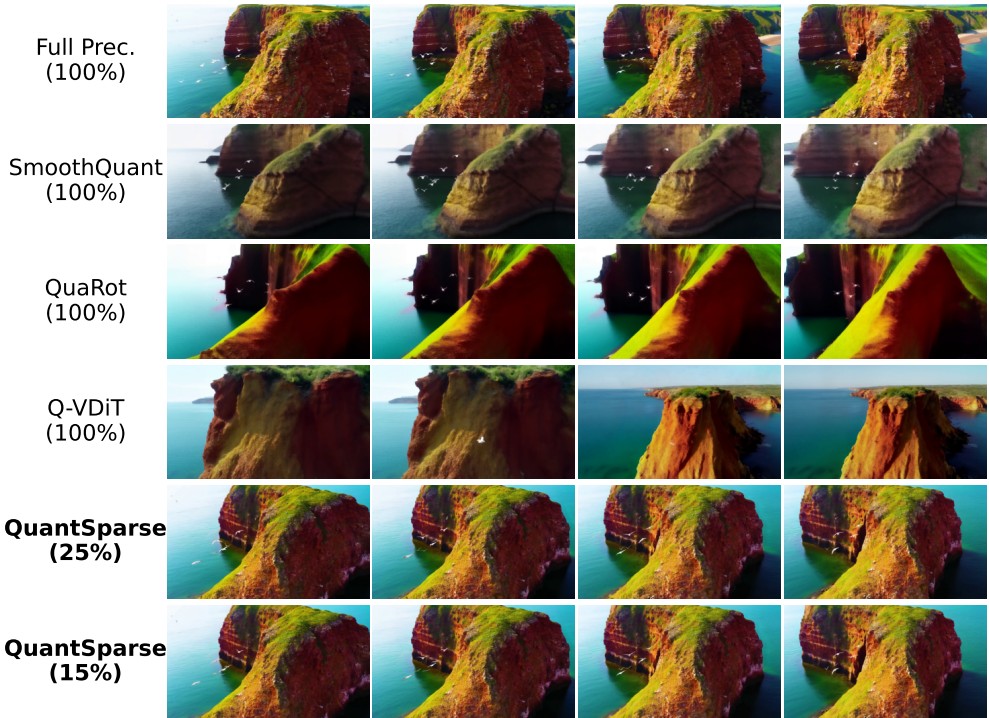

Figure 14: HunyuanVideo-13B results.

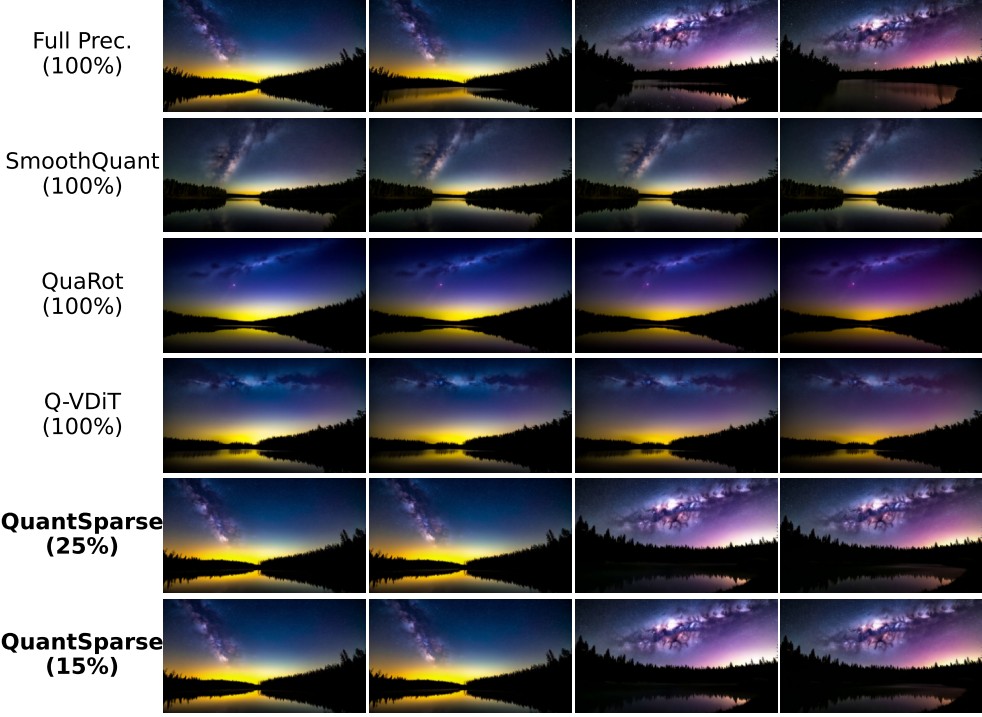

Figure 15: HunyuanVideo-13B results.

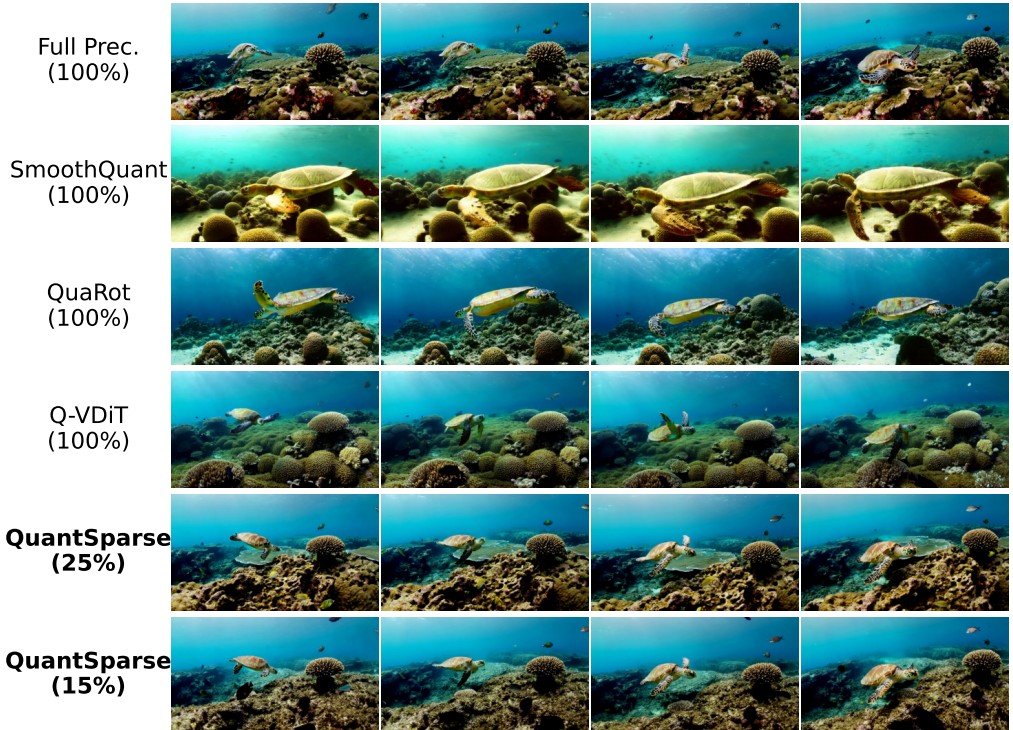

Figure 16: Wan2.1-14B results.

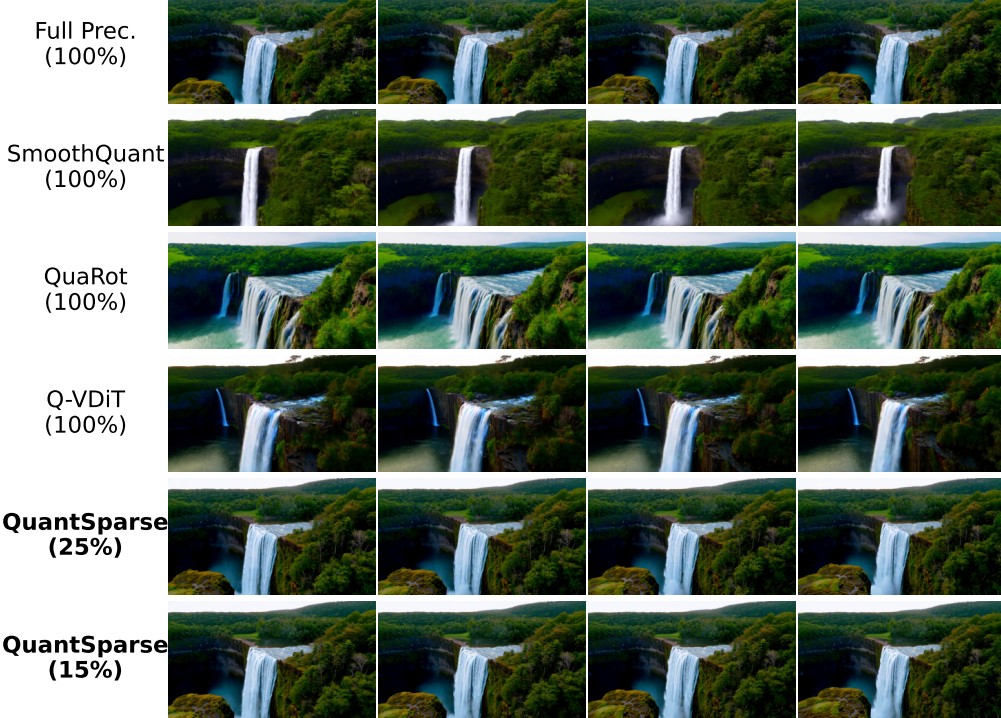

Figure 17: Wan2.1-14B results.

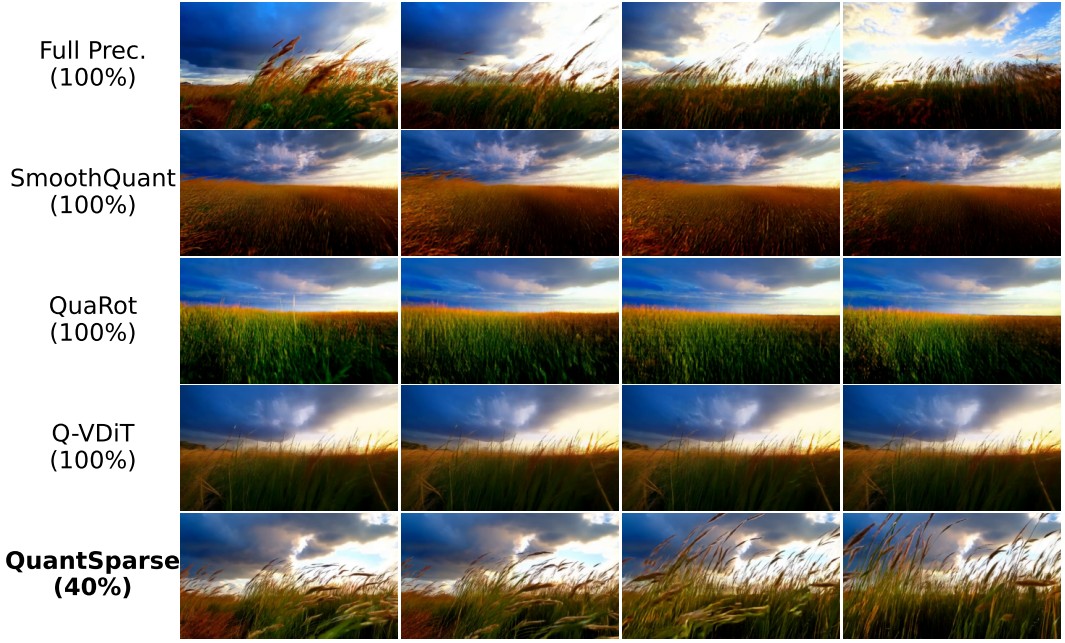

Figure 18: Wan2.1-1.3B results.

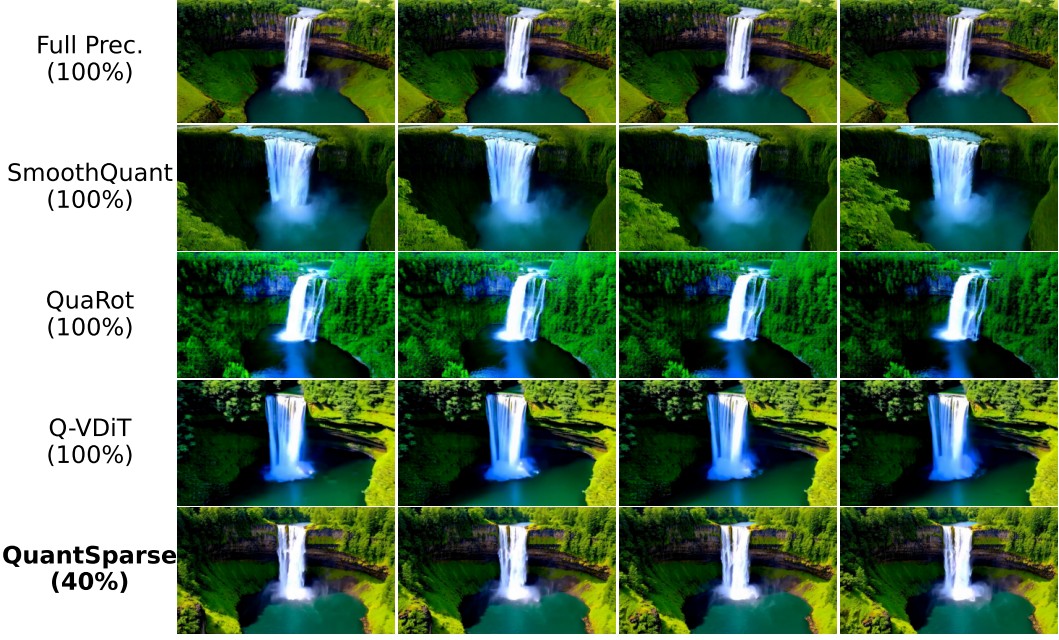

Figure 19: Wan2.1-1.3B results.

