# OpenReview forum: "QuantSparse: Comprehensively Compressing Video Diffusion Transformer with Model Quantization and Attention Sparsification"
_ICLR.cc/2026/Conference — ICLR 2026 Poster_

### Official Review · Reviewer_Ff2L · 2025-10-27

**Soundness:** 3
**Presentation:** 3
**Contribution:** 3
**Rating:** 4
**Confidence:** 3

**Summary:**

This paper proposes QuantSparse, a framework to jointly compress large video diffusion transformers using model quantization and attention sparsification. The authors argue that naively combining these two techniques leads to severe performance degradation due to "amplified attention shifts," where quantization noise exacerbates sparsity-induced information loss. To solve this, they introduce two techniques: 1. Multi-Scale Salient Attention Distillation (MSAD): A post-training quantization (PTQ) calibration method that uses a memory-efficient distillation loss. It combines (a) global guidance from downsampled attention maps to preserve coarse structure and (b) local guidance focused on high-saliency tokens to preserve fine-grained details. 2. Second-Order Sparse Attention Reparameterization (SSAR): An inference-time technique that aims to recover information lost to sparsity. It posits that the second-order residual (the change in the attention residual between timesteps) is more temporally stable than the first-order residual, especially under quantization. It caches this second-order residual (projected via SVD) to apply a more accurate correction at subsequent steps. Experiments on large models (Hunyuan-13B, Wan2.1-14B) show that QuantSparse achieves significant storage (3.8x) and latency (1.88x) reductions while claiming to maintain or even exceed the quality of the full-precision models.

**Strengths:**

1. Compressing large-scale video generation models is a critical bottleneck for their practical deployment. The paper identifies the interaction between quantization and sparsification as the key challenge, which is a more nuanced and accurate problem statement than addressing each in isolation.
2. The two proposed components are well-motivated.
3. The method achieves good performance, substantially outperforming existing quantization-only (Q-VDIT) and sparsification-only (SVG) methods, as well as their naive combinations (e.g., Q-VDIT+SVG).
4. The authors test on multiple, very large models and use a comprehensive suite of evaluation metrics, including VBench and standard video quality metrics (PSNR, SSIM, LPIPS, VQA).

**Weaknesses:**

1. Unexplained "Better-than-FP" Results: The most significant concern is that the compressed model (QuantSparse) outperforms the 16-bit Full Precision (FP) baseline on key quality metrics.

In Table 5 (Hunyuan-13B, W6A6), QuantSparse (15%) achieves a VQA score of 82.26, while the FP model scores 81.23.

In Table 6 (Wan2.1-14B, W4A8), QuantSparse (15%) achieves an "Imaging Quality" score of 63.81, while the FP model scores 63.38.

While quantization can occasionally act as a regularizer, such outperformance on large-scale generative models is an extraordinary claim that requires an explanation. The authors do not acknowledge, investigate, or explain this phenomenon. This could suggest an undertrained FP baseline, a flaw in the evaluation metrics, or a fundamental misunderstanding of the model's behavior. Without a thorough discussion, these results cast doubt on the validity of the entire evaluation.

2. Inconsistent/Confusing Ablation Study: The paper's narrative motivates MSAD as the primary solution to the core "amplified attention shift" problem. However, the ablation study in Table 3 seems to tell a different story and is presented in a highly confusing manner.

Assuming the "Cache Analysis" block's "None" row (68.00 VQA) represents an MSAD-only model (with no cache) and the "Distillation Analysis" block's "None" row (81.92 VQA) represents an SSAR-only model (with no distillation).

If this interpretation is correct, it implies that SSAR-only (81.92 VQA) provides a vastly larger performance gain over the "None (Cache)" baseline (68.00 VQA) than MSAD-only does. In fact, it suggests SSAR is the dominant contribution, and MSAD provides a smaller, secondary boost (from 81.92 VQA to 91.98 VQA).

This directly contradicts the paper's core motivation, which focuses on the quantization/sparsity interaction (MSAD's job). The authors must clarify the baselines in Table 3 and reconcile this apparent contradiction.

3. The SSAR method introduces a key hyperparameter: the "cache-refreshing interval" (set to 5). This parameter dictates the trade-off between the accuracy of the cached residual and the computational overhead of re-calculating the full-attention map. This is a critical parameter that is not ablated in the paper or appendix.

Minor:
4. I wish the author could reorganize the tables to combine the results table with the Efficiency comparison table to better illustrate.

**Questions:**

See weakness. I hope the authors could clarify these issues, and I am willing to reevaluate based on the response.

---

> ### Author Response · Authors · 2025-11-20
>
> We sincerely appreciate your careful review and valuable insights. Your questions help us clarify critical ambiguities in the ablation study and strengthen the rigor of our work. Below are detailed responses to each concern,
>
> >**Q1:** Explanation about slightly outperforms the full-precision (FP) baseline.
>
> Thank you for highlighting this phenomenon. We have carefully reflected on this and provide our interpretation below.
> 1. **MSAD as Structure- and Saliency-Aware Alignment (Beyond Light Fine-Tuning)**.
> MSAD does more than lightly fine-tune the model: it explicitly encourages the compressed model to **concentrate on task-critical tokens** and **suppress redundant or noisy tokens** present in the FP model. Our saliency analysis (Figure 3(a), Figures 6 and 7) reveals that large video DiTs exhibit a **heavy-tailed saliency distribution**, i.e., many tokens receive very low attention in the FP model. Such low-saliency tokens may contribute little or even introduce mild noise. By combining global structural guidance and local top-$k$ salient supervision, MSAD effectively re-weights the model to focus on meaningful structures, which can slightly improve certain metrics.
> 2. **SSAR as a Temporal EMA-Like Mechanism.**
> SSAR’s design is analogous to an **EMA (Exponential Moving Average)** mechanism [1] that aggregates temporally stable information. By leveraging second-order residuals across timesteps, SSAR stabilizes the sparse attention correction and mitigates random quantization noise and sparsity-induced errors. This can lead to **smoother and more temporally consistent outputs** over the multi-step denoising process, sometimes outperforming the original FP checkpoint on specific metrics.
> 3. **Consistency with Prior Works.**
> The phenomenon that carefully compressed models slightly outperform FP baselines has also been reported in prior video DiT compression works:
>     * Vidit-Q [2]: Under W6A6, the compressed model yields a higher VQA score$\uparrow$ (64.45) compared to the FP model (63.40) on Open-SORA [3] model.
>     * Q-VDiT [4]: On Latte [5] model, the compressed model achieves a $\Delta$ FLOW Score$\downarrow$ (2.70) better than the FP baseline (3.37) after temporal distillation.
> These consistent observations across different methods and models suggest that **targeted compression plus alignment can act as an effective regularizer** for large video DiTs.
> 4. **Multi-Dimensional Nature of Video Quality.**
> We stress that video generation quality is **multi-faceted**: VQA measures common video distortions; PSNR/SSIM quantify pixel-level similarity; VBench captures temporal and semantic aspects, etc. QuantSparse occasionally yields **slight improvements on some metrics**, but **does not uniformly surpass FP** across all dimensions. The main goal is to achieve **near-lossless quality while obtaining substantial compression and acceleration**, which we believe is a practically desirable Pareto trade-off rather than an indication of an underpowered FP baseline.
>
> We have added these discussions to the revised manuscript to clarify why “better-than-FP” results can occur and why they are consistent with prior literature.
>
> [1]. Tarvainen A, Valpola H. Mean teachers are better role models: Weight-averaged consistency targets improve semi-supervised deep learning results. NeurIPS 2017.
>
> [2]. Zhao T, et al. Vidit-q: Efficient and accurate quantization of diffusion transformers for image and video generation. ICLR 2025.
>
> [3]. Zheng Z, et al. Open-sora: Democratizing efficient video production for all. Arxiv 2024.
>
> [4]. Feng W, et al. Q-VDiT: Towards Accurate Quantization and Distillation of Video-Generation Diffusion Transformers. ICML 2025.
>
> [5]. Ma X, et al. Latte: Latent diffusion transformer for video generation. TMLR 2024.

---

> ### Author Response · Authors · 2025-11-20
>
> >**Q2:** Questions about the ablation study results.
>
> We appreciate the reviewer's in-depth examination of our ablation study. We clarify both the motivation and the interpretation of Table 3.
> 1. **Complementary Roles of MSAD and SSAR.**
> Both MSAD and SSAR are explicitly designed to mitigate the** “amplified attention shift”** caused by the joint use of quantization and sparsification:
>     * **MSAD** addresses **quantization-induced bias** via attention distillation, learning parameters to align the compressed model’s attention with the FP model.
>     * **SSAR** compensates for **sparsity-induced information loss** during inference by dynamically reconstructing attention using second-order residuals.
> Thus, from a conceptual standpoint, they are **equally important and complementary**: MSAD tackles the calibration side, while SSAR tackles the inference-time reconstruction side. We now clarify this explicitly in **Section 3.3** (Line 254 in the revised version).
> 2. **Balanced Contributions Across Metrics.**
> We agree that focusing solely on VQA may suggest SSAR provides the larger gain. However, **video quality is multi-dimensional**. Across metrics like PSNR and SSIM, the contributions of MSAD and SSAR are more balanced:
>     * For example, MSAD improves PSNR by +4.37, while SSAR improves PSNR by +4.56.
>     * Similarly, both significantly boost SSIM compared to the baseline.
> On VQA, SSAR indeed contributes a larger marginal improvement, which may be due to its stronger effectiveness in suppressing frame-level noise during inference. However, **MSAD also brings a large improvement**, raising VQA from **81.92 to 91.98 (+10.06)**. We now emphasize this point to avoid the misinterpretation that MSAD is secondary.
> 3. **Cache-Based Methods: Advantage from SSAR Rather than “Any Cache”.**
> It is also important to distinguish our **second-order SSAR** from **simple first-order cache-based methods**. The substantial gain seen in the “cache analysis” block of Table 3 is **not** solely due to using a cache, but due to the **specific design of SSAR** (second-order residual + SVD projection). We have clarified in the text that **naïve first-order cache methods perform significantly worse**, highlighting the unique effectiveness of SSAR.
>
> In summary, both MSAD and SSAR are **key components** of QuantSparse, motivated from complementary perspectives and providing substantial gains across different metrics. We have revised Section 3 and the ablation discussion to make this balanced view more explicit.
>
> >**Q3:** Ablation study on "cache-refreshing interval".
>
> We thank you for this important suggestion. We have added a detailed ablation study of the **cache-refreshing interval** in the revised manuscript, conducted on **Wan2.1-14B** under W4A8 with 15% density:
>
> |Method|Interval|PSNR$\uparrow$|SSIM$\uparrow$|LPIPS$\downarrow$|Speedup|
> |-|-|-|-|-|-|
> |Wan2.1-14B|-|-|-|-|1.00$\times$|
> |QuantSparse|3|18.86|0.631|0.243|1.65$\times$|
> |QuantSparse|4|18.48|0.617|0.260|1.69$\times$|
> |QuantSparse|5|18.22|0.605|0.272|1.74$\times$|
> |QuantSparse|6|17.72|0.566|0.321|1.76$\times$|
>
> We observe a clear **trade-off:**
> * Shorter intervals (e.g., 3 or 4) yield **higher PSNR/SSIM and lower LPIPS**, but **slightly reduce speedup** (1.65×–1.69×).
> * Longer intervals (e.g., 6) maximize speedup (1.76×) but cause noticeable performance degradation (PSNR drops to 17.72, SSIM to 0.566).
>
> We choose **interval = 5** in the main experiments because it provides a **good compromise** between performance and speed (PSNR 18.22, speedup 1.74×). We also note that all tested intervals produce **reasonable results and non-trivial acceleration**, so the method is not overly sensitive to this hyperparameter.
>
> This ablation and justification have been added to **Section 4.3 Table 5** in the revised manuscript.
>
> >**Q4:** Reorganize the tables to combine the results table with the efficiency comparison table.
>
> Thank you for this helpful suggestion to improve readability. In the revised manuscript:
> * We **reorganize the main results table (Table 2)** to jointly present **quality metrics and inference latency/speedup**, giving a more holistic view of the trade-offs.
> * We retain **Table 4** as a dedicated efficiency comparison to disentangle the effects of **quantization-only** vs. **sparsification-only** baselines and to highlight the advantage of combining both within QuantSparse.
>
> We believe this reorganization makes the experimental results more intuitive and easier to interpret.

---

> > ### Comment · Reviewer_Ff2L · 2025-11-22
> >
> > Thank you for the thorough revision and rebuttal; my questions have been addressed, and I will raise my score to 6.

---

> > > ### Author Response · Authors · 2025-11-23
> > >
> > > Dear reviewer,
> > >
> > > We sincerely appreciate your time and constructive feedback throughout the review process. We are delighted to hear that our rebuttal has addressed your concerns and that you now recommend acceptance. Your insightful comments have significantly strengthened our paper, and we are grateful for your valuable contribution to improving our work.
> > >
> > > Best wishes,
> > >
> > > All authors

---

### Official Review · Reviewer_N8bA · 2025-10-29

**Soundness:** 3
**Presentation:** 3
**Contribution:** 3
**Rating:** 6
**Confidence:** 3

**Summary:**

This paper presents **QuantSparse**, a unified framework for compressing large video diffusion transformers (DiTs) by combining model quantization and attention sparsification. The core problem it addresses is that naively integrating these two compression techniques leads to compounded errors and severe performance degradation. The authors propose a two-part solution: 1) **Multi-Scale Salient Attention Distillation** (MSAD), a post-training alignment method to mitigate quantization-induced bias , and 2) **Second-Order Sparse Attention Reparameterization** (SSAR), an inference-time technique to recover information lost to sparsity. The method achieves impressive results, demonstrating significant reductions in model storage and latency while maintaining the generation quality of the original full-precision models.

**Strengths:**

1. **Alignment of Non-Trivial Techniques:** The main strength of this work is the **synergistic alignment** of two orthogonal and error-introducing acceleration techniques (quantization and sparsification). The paper correctly identifies that the key challenge lies in managing the compounded errors, and it proposes a dedicated solution for each error source (MSAD for quantization bias, SSAR for sparsity loss) within a single, unified pipeline. This method could also be combined with other methods like SageAttention and TeaCache.

2.  **Excellent Empirical Results:** The method delivers on its promise. The experiments show a significant reduction in model size (e.g., **3.80x** for Wan2.1-14B) and inference latency (e.g., **1.74x** to **1.88x**) across multiple large-scale models. Crucially, this efficiency gain comes with almost no loss—and in some cases, a slight improvement—in video quality metrics compared to the full-precision baselines.

3.  **Good Presentation:** The paper is well-written and clearly presented. The problem statement is well-motivated, and figures like Figure 2, 3, and 4 effectively illustrate the proposed mechanisms and the empirical phenomena that motivate them (e.g., saliency distribution, residual stability).

**Weaknesses:**

1.  **Complex, Engineering-Heavy Pipeline:** One possible weakness is the **high complexity of the overall system**. The QuantSparse pipeline is highly engineered and involves multiple, intricate components:
    * MSAD is a two-branch distillation process requiring both global average pooling and local top-k salient token selection.
    * SSAR is a multi-stage inference process involving caching, calculating second-order residuals, and then further applying SVD-based projection to denoise these residuals.

    This makes the solution appear less like a simple, generalizable algorithm and more like a complex, bespoke system, which may hinder its broader adoption.

2. **Post-Training Alignment Requirement:** The method relies on a post-training calibration/alignment step (MSAD). This adds overhead compared to a zero-shot compression method, though this is also necessary for PTQ. This alignment step allows the authors to identify and exploit key internal patterns of the DiT, such as the heavy-tailed token saliency distribution shown in Figure 3a, which is a useful finding.

3.  **Theoretical Rigor:** The main weakness lies in the theoretical justification, which is **not tight**. The "theory" section feels more like a *post-hoc* justification for the engineering choices rather than a *first-principles* derivation.
    * **Proposition 3.3 as an Assumption:** The entire SSAR method hinges on **Proposition 3.3**, which states that the second-order residual is more temporally stable than the first-order. This is the most critical claim, yet it is presented as a "proposition" when it is, in fact, an **empirical assumption** justified only by Figure 4a. The paper attributes this to quantization noise being a "slow-varying stochastic process", which is another unproven assumption.
    * **Tautological Theorem:** **Theorem 3.4** is presented as a theoretical guarantee but is effectively **tautological (a circular argument)**. The proof in Appendix A essentially demonstrates that *if* one assumes Proposition 3.3 is true (i.e., the second-order error is smaller), *then* the second-order approximation method has a smaller expected error. This theorem does not provide any new insight beyond restating its own assumption.

**Questions:**

I find it surprising that the compressed model not only matches but, in several instances, **slightly improves upon the full-precision (FP) baseline** on key video quality metrics (e.g., VQA scores in Table 1 2).

This is an interesting result. Do you have insights on why this occurs?

* Is this improvement simply an artifact of the post-training alignment (MSAD), which may act as a light fine-tuning step, effectively regularizing the model or cleaning up minor noise from the original FP checkpoint?
* Or, does this hint at a more fundamental property of these large video DiTs? For example, could it be that the models are **significantly over-parameterized**, and that the aggressive, aligned compression from QuantSparse acts as a powerful regularizer, forcing the model to rely on its most robust features and leading to a more generalizable output?
* Or is it the metric VQA itself that has some special property for DiTs?

---

> ### Author Response · Authors · 2025-11-20
>
> We greatly appreciate your positive feedback on the motivation, empirical results, and presentation of our work. Below we clarify your concerns in more depth.
>
> >**Q1:** Complexity and generalizability of our proposed pipeline.
>
> We appreciate the reviewer’s concern about the pipeline complexity and would like to clarify that the QuantSparse design goal is to **balance effectiveness, practicality, and generalizability**, and our ablations and overhead analysis support that the pipeline is **both stable and lightweight** in practice.
>
> 1. **Robustness to Hyperparameters.**
> Our components show strong insensitivity to hyperparameters:
>     * For **MSAD**, varying the pooling stride $s$ (64/128/256) and salient token $k$ (128/256/512) leads to very small performance variations (VQA fluctuates within 1.2, PSNR within 0.3).
>     * For the distillation loss weights $\lambda_{global}$ and $\lambda_{local}$, multiple settings (5e-3 / 1e-4 / 5e-4) consistently outperform the baseline without requiring fine-grained tuning.
> 2. **Low Computational and Memory Overhead.**
> As detailed in **Table 14 (calibration resource)** and **Table 15 (inference resource)**, the additional overhead is small:
>     * MSAD adds only **0.3–0.8%** GPU memory and **0.2–1.6%** time overhead over naïve PTQ.
>     * SSAR introduces merely **7–11%** memory and **0.2–0.8%** time overhead compared to non-reparameterized sparse attention.
>     These overheads are negligible relative to the overall cost of large video DiTs.
> 3. **Strong Cross-Model Generalizability.**
> We validate QuantSparse on **three large-scale video diffusion models**: Wan2.1-1.3B, Wan2.1-14B, and HunyuanVideo-13B (Tables 1, 2, 6, 7). In all cases, QuantSparse outperforms both **quantization-only** and **sparsification-only** baselines and their naïve combinations. For instance, on Wan2.1-14B with 15% attention density, QuantSparse achieves a VQA score of **90.73**, comparable to the FP model’s **90.79**, while delivering substantial compression and acceleration. This demonstrates that QuantSparse is not a bespoke design for a single model, but a **generalizable framework** for video DiT compression.
>
> >**Q2:** Post-training alignment costs.
>
> We agree that post-training calibration introduces additional cost compared to purely zero-shot methods. However, we emphasize that **QuantSparse’s alignment step (MSAD) is extremely lightweight and yields significant performance gains.**
> * As reported in **Table 14**, the calibration overhead remains modest even for large models:
>     * For **Wan2.1-14B**, the full QuantSparse calibration takes only **2.60 hours**.
>     * For **Wan2.1-1.3B**, calibration is even cheaper, requiring only **0.64 hours**.
> * Despite this low cost, MSAD leads to **substantial performance improvements**:
>     * On Wan2.1-14B under W4A8 and 25% density, QuantSparse improves PSNR to **18.72**, which is **+4.91** higher than the zero-shot QuaRot baseline (13.79) and **+2.87** higher than Q-VDiT (15.85).
>     * Relative to pure quantization methods, QuantSparse maintains superior quality while achieving **3.80× storage compression** and **1.74× inference acceleration** (Table 4).
>
> Thus, the small one-time calibration cost is amply justified by the large, persistent gains in both quality and efficiency.
>
> >**Q3:** Question about empirical assumption.
>
> We appreciate your attention to theoretical rigor and clarify the roles of **Proposition 3.3** and **Theorem 3.4**.
> * **Empirical Foundation of Proposition 3.3.**
> Proposition 3.3 states that second-order residuals exhibit lower temporal variance than first-order ones, motivating our second-order residual-based SSAR design. This is **not assumed in isolation**; it is backed by extensive empirical evidence:
>     * In **Figure 4** and **Figure 10**, we visualize residual temporal differences across **five transformer blocks of HunyuanVideo-13B**, consistently showing that second-order residuals have significantly smaller variance than first-order ones.
>     * Similar phenomena are also observed on Wan2.1, as reported in **Table 11**, indicating that this property holds across different models and layers.
> * **Role of Theorem 3.4.**
> Theorem 3.4 builds a **direct link between the temporal variance phenomenon and the expected attention error bound:**
> $$\mathbb{E}_ t \left\| A_ {full}^{(t)} - \tilde{A}_ {s,q}^{(t)} \right\|_ F \leq \mathbb{E}_ t \left\| A_ {full}^{(t)} - \hat{A}_ {s,q}^{(t)} \right\|_ F$$
> Given the empirically established variance property (Proposition 3.3), Theorem 3.4 **strictly proves second-order residual-based SSAR yields a tighter attention error bound** than first-order approaches. While the theorem builds on an empirically observed property, it provides a principled bridge between this property and the design choice of SSAR.
>
> We have revised the text to more clearly distinguish between **empirical observations** and **theoretical consequences**, and to make the logical dependence explicit.

---

> ### Author Response · Authors · 2025-11-20
>
> >**Q4:** Explanation about slightly outperforms the full-precision (FP) baseline.
>
> Thank you for highlighting this phenomenon. We have carefully reflected on this and provide our interpretation below.
> 1. **MSAD as Structure- and Saliency-Aware Alignment (Beyond Light Fine-Tuning)**.
> MSAD does more than lightly fine-tune the model: it explicitly encourages the compressed model to **concentrate on task-critical tokens** and **suppress redundant or noisy tokens** present in the FP model. Our saliency analysis (Figure 3(a), Figures 6 and 7) reveals that large video DiTs exhibit a **heavy-tailed saliency distribution**, i.e., many tokens receive very low attention in the FP model. Such low-saliency tokens may contribute little or even introduce mild noise. By combining global structural guidance and local top-$k$ salient supervision, MSAD effectively re-weights the model to focus on meaningful structures, which can slightly improve certain metrics.
> 2. **SSAR as a Temporal EMA-Like Mechanism.**
> SSAR’s design is analogous to an **EMA (Exponential Moving Average)** mechanism [1] that aggregates temporally stable information. By leveraging second-order residuals across timesteps, SSAR stabilizes the sparse attention correction and mitigates random quantization noise and sparsity-induced errors. This can lead to **smoother and more temporally consistent outputs** over the multi-step denoising process, sometimes outperforming the original FP checkpoint on specific metrics.
> 3. **Consistency with Prior Works.**
> The phenomenon that carefully compressed models slightly outperform FP baselines has also been reported in prior video DiT compression works:
>     * Vidit-Q [2]: Under W6A6, the compressed model yields a higher VQA score$\uparrow$ (64.45) compared to the FP model (63.40) on Open-SORA [3] model.
>     * Q-VDiT [4]: On Latte [5] model, the compressed model achieves a $\Delta$ FLOW Score$\downarrow$ (2.70) better than the FP baseline (3.37) after temporal distillation.
> These consistent observations across different methods and models suggest that **targeted compression plus alignment can act as an effective regularizer** for large video DiTs.
> 4. **Multi-Dimensional Nature of Video Quality.**
> We stress that video generation quality is **multi-faceted**: VQA measures common video distortions; PSNR/SSIM quantify pixel-level similarity; VBench captures temporal and semantic aspects, etc. QuantSparse occasionally yields **slight improvements on some metrics**, but **does not uniformly surpass FP** across all dimensions. The main goal is to achieve **near-lossless quality while obtaining substantial compression and acceleration**, which we believe is a practically desirable Pareto trade-off rather than an indication of an underpowered FP baseline.
>
> We have added these discussions to the revised manuscript to clarify why “better-than-FP” results can occur and why they are consistent with prior literature.
>
> [1]. Tarvainen A, Valpola H. Mean teachers are better role models: Weight-averaged consistency targets improve semi-supervised deep learning results. NeurIPS 2017.
>
> [2]. Zhao T, et al. Vidit-q: Efficient and accurate quantization of diffusion transformers for image and video generation. ICLR 2025.
>
> [3]. Zheng Z, et al. Open-sora: Democratizing efficient video production for all. Arxiv 2024.
>
> [4]. Feng W, et al. Q-VDiT: Towards Accurate Quantization and Distillation of Video-Generation Diffusion Transformers. ICML 2025.
>
> [5]. Ma X, et al. Latte: Latent diffusion transformer for video generation. TMLR 2024.

---

### Official Review · Reviewer_5UCB · 2025-10-30

**Soundness:** 3
**Presentation:** 3
**Contribution:** 3
**Rating:** 6
**Confidence:** 3

**Summary:**

This paper proposes QuantSparse, a unified framework that jointly applies model quantization and attention sparsification to compress video diffusion transformers efficiently. To address the performance drop caused by their naive combination, the authors introduce: Multi-Scale Salient Attention Distillation (MSAD) for aligning quantized and sparse attention using global–local supervision, and Second-Order Sparse Attention Reparameterization (SSAR) for recovering information lost under sparsity via temporally stable residuals.
Experiments on HunyuanVideo and Wan2.1 models show that QuantSparse achieves up to 3.8× compression and 1.7× speedup with minimal quality loss, outperforming prior quantization and sparsification baselines.

**Strengths:**

The paper demonstrates strong methodological innovation by proposing the first unified framework that integrates model quantization and attention sparsification, along with a systematic analysis of the “amplified attention shift” problem, showcasing both theoretical depth and engineering value. The proposed MSAD and SSAR modules effectively mitigate performance degradation from the perspectives of distillation calibration and dynamic reconstruction, balancing global and local features with good generality and scalability. Extensive experiments on multiple video generation models validate the approach, showing that QuantSparse significantly outperforms existing baselines in metrics such as PSNR, VQA, and LPIPS, while achieving notable model compression and inference acceleration.

**Weaknesses:**

1. Figure 2 serves as the core illustration of the QuantSparse framework; however, the manuscript does not provide sufficient explanation of each module’s function, their interrelations, or the meaning of the symbols used. It is recommended that the authors include detailed annotations and descriptions of key steps and variables in the main text or appendix to enhance the clarity and reproducibility of the method.

2. The main objective of this paper is to achieve efficient video diffusion model deployment on consumer-grade GPUs through model quantization. Therefore, the specific experimental hardware configurations (e.g., GPU model, memory capacity, and system environment) are crucial for assessing the practical feasibility of the proposed method. The authors are encouraged to explicitly include these hardware details in the main text.

3. In the Introduction, the authors state that directly combining quantization and sparsification causes an “amplified attention shift,” leading to performance degradation. However, only theoretical analysis is provided without corresponding experimental data or visual evidence. It is suggested that the authors include quantitative comparisons or visualizations to more convincingly demonstrate the existence and impact of this phenomenon.

4. The experiments employ a 15% attention sparsity ratio, yet the rationale or tuning process behind this choice is not explained. The current setup appears to be empirical and lacks systematic hyperparameter analysis. The authors are advised to clarify the basis for selecting this sparsity level and to provide comparative experiments, such as results at 10%, 20%, and 30% sparsity levels, showing the trade-off between model performance and inference acceleration in the main text or appendix to strengthen the persuasiveness of the results.

**Questions:**

For more details, see weakness

---

> ### Author Response · Authors · 2025-11-20
>
> We are grateful for your positive assessment of our contributions and for your helpful suggestions. We respond to your questions as follows.
>
> >**Q1:** Clarifications on Figure 2 (Core Framework Illustration).
>
> We agree that Figure 2 is central to understanding QuantSparse and have enhanced its clarity in the revised manuscript.
>
> * We now provide **more structured and detailed descriptions** in the Figure 2 caption, clearly annotating each module (MSAD global branch, MSAD local branch, SSAR, and their interactions).
> * We add an **overall pipeline description** in Section 3.4, which walks the reader through the end-to-end procedure of QuantSparse.
> * We also include a **supplementary algorithm (Algorithm 1)** in the appendix that explicitly summarizes the step-by-step procedure of our method.
>
> These updates make the core framework and information flow of QuantSparse significantly more transparent and reproducible.
>
> >**Q2:** Specific experimental hardware configurations.
>
>
> Thank you for emphasizing the importance of hardware details for assessing practical feasibility. In the revised manuscript, we have explicitly added the hardware and system environment in **Section 4.4 (Efficiency Analysis)**:
>
> "All the experiments are conducted on a single NVIDIA A800 80G GPU with CUDA 12.4. We use CUTLASS on top of PyTorch for performing INT matrix multiplication."
>
> This clarifies the resource assumptions underlying our efficiency and deployment claims.
>
> >**Q3:** Experimental & Visual Evidence for "Amplified Attention Shift".
>
> We appreciate this suggestion and have strengthened the empirical evidence for the “amplified attention shift” phenomenon from both **visual** and **quantitative** perspectives.
>
> * **Visualizations**
>     * In the main paper, Figure 3(b) shows that the naïve combination of quantization and sparsification produces blurred and misaligned attention, while Figure 3\(c) demonstrates that MSAD realigns attention to FP structural and salient regions.
>     * We further provide **four additional sets of attention map comparisons** **in Appendix Section F, Figures 8 and 9**, contrasting FP attention, naïve quantized+sparse attention, and QuantSparse attention. The corresponding **MSE metrics** (without vs. with attention distillation) are also plotted, showing that MSAD consistently reduces attention error.
> * **Quantitative Validation**
>     To directly quantify the effect of combining quantization and sparsification, we measure the **attention Mean Squared Error (MSE)** on Wan2.1-1.3B [1] under W4A8 with 40% attention density over 1000 random samples:
>
> |Method|Attention Shift (MSE$\downarrow$)|
> |-|-|
> |Quantization (QuaRot [2])|0.216|
> |Sparsification (SVG [3])|0.134|
> |**Quantization+Sparsification**|**0.685**|
>
> The joint attention shift (0.685) is **much larger** than the sum of individual effects, confirming that quantization and sparsification interact to produce **amplified attention shift**, which directly motivates our design of MSAD and SSAR.
>
> These results and figures have been incorporated into **Appendix Section F (Table 8 and Figures 8–9)** in the revised manuscript.
>
> [1]. Wan T, et al. Wan: Open and advanced large-scale video generative models. Arxiv 2025.
>
> [2]. Ashkboos, et al. QuaRot: Outlier-Free 4-Bit Inference in Rotated LLMs. NeurIPS 2024.
>
> [3]. Xi H, et al. Sparse videogen: Accelerating video diffusion transformers with spatial-temporal sparsity. ICML 2025.
>
> >**Q4:** Systematic Analysis of Sparsity Ratio.
>
> Thank you for pointing out the need for a systematic study of attention density. In the revised manuscript, we add a comprehensive ablation on **Wan2.1-14B [1]** under W4A8 quantization, varying the attention density and reporting both quality and speed:
>
> |Method|Density$\downarrow$|PSNR$\uparrow$|SSIM$\uparrow$|LPIPS$\downarrow$|Speedup|
> |-|-|-|-|-|-|
> |Wan2.1-14B|100%|-|-|-|1.00$\times$|
> |QuantSparse|25%|18.72|0.630|0.240|1.55$\times$|
> |QuantSparse|20%|18.45|0.622|0.252|1.63$\times$|
> |QuantSparse|15%|18.22|0.605|0.272|1.74$\times$|
> |QuantSparse|10%|17.73|0.589|0.288|1.80$\times$|
>
> A 25% density achieves a **good balance** between speed and quality (1.55× speedup with PSNR 18.72), while 15% density further improves speedup to 1.74× with still acceptable quality (PSNR 18.22). Based on this trade-off:
> * We adopt **25% density** as a high-quality setting with strong fidelity and noticeable acceleration.
> * We adopt **15% density** to demonstrate the potential for more aggressive acceleration with limited quality degradation.
>
> These results and our rationale have been added to **Section 4.3 Table 5** in the revised manuscript.
>
> [1]. Wan T, et al. Wan: Open and advanced large-scale video generative models. Arxiv 2025.

---

### Official Review · Reviewer_TKrA · 2025-11-01

**Soundness:** 3
**Presentation:** 3
**Contribution:** 3
**Rating:** 6
**Confidence:** 3

**Summary:**

In this work, QuantSparse is proposed for compressing video diffusion transformers, with both attention sparsification and quantization applied. The authors point out that, when both techniques are naïvely applied, the error introduced by quantization will induce attention shift, which negatively impacts model performance. In QuantSparse, MSAD is used to efficiently perform attention distillation, leveraging both global and local guidance, and SSAR is proposed to exploit the temporal stability of the second-order residual to enhance sparse attention. The results show that diffusion transformers compressed by the proposed method can achieve SOTA performance while reducing both storage and latency simultaneously.

**Strengths:**

- The paper provides a detailed analysis of the problem caused by the combined use of attention sparsity and quantization. The proposed method is rational, easy to implement, and has low overhead, while demonstrating promising performance.
- The results are impressive. It improves the inference efficiency of SOTA video generation models with minimal degradation in generation quality.

**Weaknesses:**

- There are some clarity issues in the paper. Some symbols are reused for different hyperparameters. For example, the pooling stride is denoted as $s$ in Eq. (6) and the SVD rank is denoted as $r$ in Section 4.3, but the pooling stride becomes $r$ in Section H. See also the questions below.

**Questions:**

- Is the proposed method applicable to other transformer models that are not designed for video generation?
- How does the attention distillation perform when using full attention as the target?
- Could you clarify the calculation of the saliency? In Eq. (7), it is written as $s_{i}=\sum_{j}A_{j, i}$, but in the figure 2 it is $\sum_{h, i}A_{h, i, j}$.
- Given the large pooling stride used, what are the attention maps resolution? How do they impact the distillation time?

---

> ### Author Response · Authors · 2025-11-20
>
> We sincerely thank you for your positive assessment of our work and your constructive suggestions. We address your comments in detail below.
>
> >**Q1:** Clarity Issues: Symbol Uniformity & Saliency Calculation.
>
> Thank you for your valuable suggestions and we have carefully revised the manuscript to improve notation consistency and clarify the saliency computation.
>
> * **Symbol Uniformity:** We have standardized the notation as follows:
>     * Pooling stride (used in global distillation of MSAD) is uniformly denoted as $s$ (consistent in Eq. (6) and Appendix Section H Table 10).
>     * SVD rank (used in SSAR) is retained as $r$ (Eq. (16)) to avoid confusion.
>         All conflicting symbols in the original manuscript have been corrected.
> * **Saliency Calculation:** The saliency score is computed across all attention heads, which was not clearly reflected in the original notations and led to the inconsistency between Eq. (7) and Figure 2. We have corrected this by explicitly writing:
> $$A \in \mathbb{R}^{h,L,L}, s_j = \sum_{h,i} A_{h,i,j}$$
> where $h$ denotes the attention head, $i$ denotes the key token index, and $s_j$ is the aggregated attention received by query token $j$. We have updated both Eq. (7) and Figure 2 to reflect this correction.
>
>
> >**Q2:** Applicability to Non-Video Transformer Models.
>
> We agree that demonstrating broader applicability is important. Although QuantSparse is motivated by video diffusion transformers (DiTs), it is designed as a **general framework for DiT-based generative models** rather than being restricted to video generation.
>
> To validate its generalizability, we conducted an experiment on an **image-generation DiT**: Hunyuan-DiT (1.5B parameters, targeting image generation [1]). We evaluate W4A8 quantization on the DrawBench benchmark [2] and compare against state-of-the-art DiT quantization methods QuaRot [3] and Q-VDiT [4]. The results are:
>
> |Method|Density$\downarrow$|PSNR$\uparrow$|SSIM$\uparrow$|LPIPS$\downarrow$|
> |-|-|-|-|-|
> |QuaRot|100%|17.30|0.627|0.460|
> |Q-VDiT|100%|19.23|0.658|0.347
> |**QuantSparse (Ours)**|40%|**20.34**|**0.692**|**0.289**|
>
> Even in the image-generation setting, QuantSparse surpasses strong baselines while using only **40% attention density**, confirming that our framework is not restricted to video DiTs and generalizes well to DiT-based visual generation tasks.
>
> **We have also added these results in Appendix Section K.**
>
> [1]. Li Z, et al. Hunyuan-dit: A powerful multi-resolution diffusion transformer with fine-grained chinese understanding. ArXiv 2024.
>
> [2]. Saharia C, et al. Photorealistic text-to-image diffusion models with deep language understanding. NeurIPS 2022.
>
> [3]. Ashkboos, et al. QuaRot: Outlier-Free 4-Bit Inference in Rotated LLMs. NeurIPS 2024.
>
> [4]. Feng W, et al. Q-VDiT: Towards Accurate Quantization and Distillation of Video-Generation Diffusion Transformers. ICML 2025.
>
> >**Q3:** Performance of Attention Distillation with Full Attention as Target.
>
> We appreciate this suggestion and have conducted a direct comparison between our **Multi-Scale Salient Attention Distillation (MSAD)** and **full-attention distillation** (using the entire FP attention map as the distillation target).
>
> We evaluate Wan2.1-1.3B [1] under W4A8 quantization:
>
> |Method|PSNR$\uparrow$|LPIPS$\downarrow$|Attention Memory Cost (GB)|Calibration Time (Hours)|
> |-|-|-|-|-|
> |Full Attention|15.25|0.338|6.82|1.86|
> |**MSAD**|15.22|0.338|0.14|0.64|
>
> MSAD achieves **nearly identical performance** to full-attention distillation (PSNR differs by only 0.03, LPIPS by 0.001). MSAD reduces attention memory cost by **97.9%** and calibration time by **69.3%** (where full attention maps are prohibitively large, e.g., 59.88GB for a single attention map on Wan2.1-14B as shown in Figure 3(d)).
>
> We have revised this experiment in **Appendix Section H Table 13**.
>
> [1]. Wan T, et al. Wan: Open and advanced large-scale video generative models. Arxiv 2025.

---

> ### Author Response · Authors · 2025-11-20
>
> >**Q4:** Attention Map Resolution & Impact on Distillation Time.
>
> To clarify the attention map resolution and its relationship with distillation time, we first explain the calculation basis:
>
> **The original attention map resolution** of Wan2.1-1.3B [1] is
> $$(L, L) = (17472, 17472),$$
> derived from latent tokenization of video frames. MSAD’s global distillation reduces resolution via average pooling with stride $s$, where the downsampled resolution is
> $$(\tilde{L}, \tilde{L}) = (L/s^2, L/s^2).$$
> Below are the resolution, distillation time, and performance metrics for different pooling strides on Wan2.1-1.3B (W4A8 quantization):
>
> |Pooling Stride|Resolution|PSNR$\uparrow$|LPIPS$\downarrow$|Attention Memory Cost (GB)|Calibration Time (Hours)|
> |-|-|-|-|-|-|
> |Full Attention|(17472, 17472)|15.25|0.338|6.82|1.86|
> |64|(273, 273)|15.23|0.338|0.17|0.66|
> |128|(137, 137)|15.22|0.338|0.14|0.64|
> |256|(69, 69)|15.21|0.339|0.13|0.63|
>
> As shown, the pooling stride used in MSAD shows robust performance. We choose $s=128$ as the default since it offers a favorable balance between accuracy and efficiency.
>
> We have revised this experiment in Appendix **Section H Table 13.**
>
> [1]. Wan T, et al. Wan: Open and advanced large-scale video generative models. Arxiv 2025.

---

### Author Response · Authors · 2025-11-24
**General Response**

Dear AC and Reviewers,

We sincerely thank all reviewers for their time and effort in evaluating our paper. We are pleased that the reviewers acknowledged the following contributions:

* **Strong motivation.** We systematically study the interaction between model quantization and attention sparsification in video diffusion transformers, and address the amplified attention shift phenomenon caused by naive combination. *[Reviewers TKrA, 5UCB, N8bA, and Ff2L]*
* **Tailored methodology.** The proposed MSAD and SSAR modules are explicitly designed to mitigate quantization-induced bias and sparsity-induced information loss, while remaining conceptually simple, easy to implement, and low-overhead. *[Reviewers TKrA, 5UCB, N8bA, and Ff2L]*
* **Strong performance and efficiency.** QuantSparse consistently achieves substantial storage reduction and inference acceleration on several large-scale video diffusion transformers, with minimal quality degradation and significantly outperforms existing baseline methods. *[Reviewers TKrA, 5UCB, N8bA, and Ff2L]*
* **Broad evaluation.** Our experiments cover multiple large-scale state-of-the-art video diffusion transformers and diverse metrics (e.g., PSNR, SSIM, LPIPS, VQA, VBench), demonstrating both fidelity and temporal/semantic consistency under aggressive compression. *[Reviewers 5UCB, N8bA, and Ff2L]*

We also thank all reviewers for their insightful and constructive suggestions, which helped us further improve the paper. The major revisions are summarized below:

* Clarified notation and multi-head saliency definition by unifying symbols and updating the text and figures. *[Reviewer TKrA, and 5UCB]*
* Added experiments on an image-generation DiT (Hunyuan-DiT) to show that QuantSparse also generalizes to DiT-based image models. *[Reviewers TKrA]*
* Performed systematic ablations on attention density and cache-refresh intervals to clarify quality–speed trade-offs and support our default settings. *[Reviewers 5UCB and Ff2L]*
* Added a short discussion on why the compressed model can slightly outperform FP on some metrics, linking this to saliency-aware alignment and temporal stabilization effects. *[Reviewers N8bA and Ff2L]*


In addition, we provide point-by-point responses for each reviewer and have incorporated all changes into the revised manuscript, where the modifications are highlighted in red.

Best regards,
Authors of Paper #1688

---

### Meta-Review · Area_Chair_y7zg · 2026-01-07

**Summary:**

The reviewers’ major concerns focused on the motivation and evidence for the claimed “attention shift” that arises when quantization and sparsification are combined. The authors have addressed this point well with additional analysis and experiments. Several reviewers also noted that, despite the reported memory and speed benefits, the overall method is relatively complex: the pipeline contains multiple modules, and clearer explanations are needed to preserve reproducibility and avoid the impression of a highly engineered, less “simple” solution.

I would also like to raise an additional point that was not explicitly discussed in the reviews: the absolute PSNR numbers reported in this submission are not good (around ~20 PSNR), whereas prior work on sparse attention alone (e.g., SVG, SVG2) can reach substantially higher PSNR (e.g., ~29) without quantization. This suggests that the quality gap introduced by quantization—and the extent to which it is unavoidable versus improvable—has not been fully characterized.

That said, from the perspective of diagnosing and mitigating the quantization+sparsification interaction (i.e., attention shift), the authors have done a solid job, and I recommend acceptance.

**Reviewer Concerns:**

The authors addressed well on the method clarification and the more analysis and clarification. The major outstanding concerns so far is the possibility of real deployment, it looks like the whole method lacks simplicity but I think it is effective, which is good.

**Reviewer Scores:**

The reviewer Ff2L is fully participated to the discussion and brings up the concerns about the experiments and metric clarification and the reviewer 5UCB pointed out the major motivation of attention shift was not fully demonstrated. The authors did well on addressing them. The proposed method is also effective.

---

### Decision · Program_Chairs · 2026-01-26

Accept (Poster)